# *From Images to Words:* Efficient Cross-Modal Knowledge Distillation to Language Models from Black-box Teachers

## Abstract

Knowledge distillation (KD) methods are pivotal in compressing large pre-trained language models into smaller models, ensuring computational efficiency without significantly dropping performance. Traditional KD techniques assume homogeneity in modalities between the *teacher* (source) and the *student* (target) models. On the other hand, existing multimodal knowledge distillation methods require *modality-specific pre-training* of the teacher model, which is computationally infeasible in most cases. In this paper, we introduce `ARMADA`, an efficient cross-modal knowledge distillation framework designed to transfer knowledge from large vision-language models, including black-box models, to language-only models. Unlike existing KD techniques that rely on the internal structures of multimodal teachers or require computationally expensive pre-training, `ARMADA` leverages novel alignment techniques to distil knowledge without altering the teacher model, ensuring efficiency and scalability. We empirically validate `ARMADA` on twelve natural language understanding, eight complex generative reasoning and five instruction-tuning tasks, demonstrating consistent performance improvements in large models such as DeBERTa-v2-1.4B, OPT-1.3B, LLaMA-{3B, 7B, 8B}. `ARMADA` achieves up to 3.4% improvement on language understanding tasks and 2.6% boost in generative reasoning, all without requiring expensive multimodal pre-training or fine-tuning of the teacher model. Our findings challenge conventional knowledge distillation paradigms by demonstrating that even vision-language models, despite lacking direct textual understanding, can significantly enhance language models when distilled appropriately.

## 1 Introduction

Pre-trained large language models (LLMs) (Achiam et al., 2023; Gunasekar et al., 2023; Jiang et al., 2023; Dubey et al., 2024; DeepSeek-AI et al., 2024) have demonstrated remarkable performance across various tasks in diverse domains, including commonsense and mathematical reasoning, natural language understanding, code generation and dialogue generation, showcasing their versatility and effectiveness. However, as these models continue to grow in size and complexity, their computational costs become prohibitive for many applications. This has led to a surge of interest in *model compression* techniques (Buciluǎ et al., 2006; Hinton et al., 2015; Zhou et al., 2021), which aim to reduce the size of a pre-trained LLMs for computational efficiency at test-time. Hinton et al. (2015) popularized the concept of knowledge distillation (KD) for compressing large models into smaller and shallower models. The majority of the subsequently proposed KD methods (Sun et al., 2019; Zhang et al., 2018; Zhou et al., 2021) are *unimodal*, *i.e.*, predominantly work when both the teacher and student models are from the same modality. The reliance on the teacher's modality on student models prevents these methods from effectively transferring cross-modal cues and, therefore, restricts the student models from gaining a more generalized aspects of the downstream tasks.

*Cross-modal knowledge distillation* has emerged as a technique where teacher models specializing in one modality can transfer knowledge to student models in another modality. This allows the student model to assimilate concepts across different modalities, analogous to how a prompter helps a blind person perceive the world through narration. Notable cross-modal KD approaches include 'vokenization' (Tan & Bansal, 2020), which associates text with visual tokens to improve language models, and video-language KD frameworks that utilize video-based teacher models for cross-modal knowledge transfer. In a similar attempt, Tang

et al. (2021) proposed a video-language KD framework, which leverages a video-language encoder teacher model for distilling cross-modal knowledge into the student LMs. However, the effectiveness of these existing cross-modal distillation techniques relies on the large pre-training of the multimodal teacher (e.g., Tang et al. (2021) used $136M$ video clips for teacher pre-training). These computational expenses can be significantly reduced by utilizing readily available large pre-trained multimodal white-box (Rombach et al., 2022b) and black-box (Betker et al., 2023) models, which the existing cross-modal models fail to reap off.

To this end, we propose `ARMADA`‡, a cross-modal knowledge distillation framework designed to distil knowledge from any white-box or black-box text-to-vision teacher model to language-based student models. At the core of `ARMADA` is the `TS Aligner`, a module that aligns the student model with the teacher's multimodal abstraction space. `ARMADA` further utilizes novel manifold and auxiliary alignment mechanisms, which regularize students to learn different abstractions by jointly training the models with shared objectives. Unlike the existing cross-modal KD frameworks, `ARMADA` does not require student models to generate mental images but rather encourages abstract knowledge representation. This allows the framework to distil knowledge from a teacher model to different student models while maintaining efficiency and adaptability.

We empirically validate `ARMADA` on twelve natural language understanding (NLU) tasks from the GLUE (Wang et al., 2018) and SuperGLUE (Wang et al., 2019) benchmarks, as well as five commonsense and three mathematical reasoning tasks. On NLU tasks, a BERT-6L model (Devlin et al., 2018) distilled via `ARMADA` using Stable Diffusion (Rombach et al., 2022a) and Midjourney (Borji, 2022) teacher models achieves performance improvements of 3.4% and 3.2%, respectively, over its undistilled counterpart. `ARMADA` also enhances larger models (>1B parameters) such as DeBERTa-v2-xxlarge (He et al., 2020)) and OPT-1.3B (Zhang et al., 2022), by 1.4% and 1.5%, respectively. Even in zero-shot generative tasks, `ARMADA` demonstrates its efficacy by improving the performance of a pre-trained LLaMA-7B model (Touvron et al., 2023) by 0.5%, with a maximum task-specific improvement of 2.6%, proving its scalability across models of different sizes. Even with only 0.8% learnable parameters than that of the existing unimodal and multimodal KD methods, `ARMADA` is proven to equally effective in assimilating teacher supervision into student models.

The contributions of our paper are summarized below:

- **Cross-Modal Knowledge Distillation for Large Language Models.** Unlike multimodal pre-trained models such as CLIP (Radford et al., 2021) or Qwen-VL (Wang et al., 2024), in cross-modal distillation, the trainable model can access only one modality, making knowledge fusion immensely challenging. To our knowledge, `ARMADA` is the first architecture-agnostic cross-modal knowledge distillation technique from black-box teachers to language-only student models.
- **Efficient and Scalable Cross-Modal KD.** Unlike existing cross-modal KD methods, our proposed method does not require computationally expensive modality-specific pre-training of teacher models. The framework enhances student models with minimal additional learnable parameters (0.8%) than the existing methods. This efficiency, combined with its ability to work with white-box and black-box teachers, positions `ARMADA` as a practical and scalable solution for improving language models with cross-modal signals.
- **Theoretical Insights into Cross-Modal Knowledge Distillation.** We provide an analytical explanation of how cross-modal knowledge distillation works by establishing the equivalence between teacher and student manifold spaces. Although the subject being substantially least understood, our work theoretically justifies the effectiveness of cross-modal distillation for transferring abstract knowledge across modalities. This deepens our understanding of how knowledge transfer can be achieved across modalities, making the process both explainable and efficient.

## 2 Related Work

**Unimodal Knowledge Distillation.** Hinton et al. (2015) expanded the work of Buciluă et al. (2006) to introduce and popularize KD in which a shallow model tries to mimic the performance of a complex and large model through its output distribution. Instead of guiding the student through only the output layer of the teacher model, learning from intermediate features of the teacher has been proposed for better knowledge

---

‡The source code of `ARMADA` will be released upon acceptance of the paper.

transfer (Romero et al., 2014; Zagoruyko & Komodakis, 2016; Sun et al., 2019; Zhang et al., 2020). Other studies on distillation encouraged different techniques, including relation-based KD (Park et al., 2019; Chen et al., 2020) and meta KD (Pan et al., 2020; Zhou et al., 2021; Sengupta et al., 2024), for effective knowledge sharing.

**Cross-modal Knowledge Distillation.** As opposed to unimodal KD, which is primarily motivated by model compression, cross-modal knowledge distillation aims at transferring knowledge from one modality to another. This setup typically entails a pre-trained teacher from a specific modality transferring knowledge to a student from a different modality. One of the earliest studies by Gupta et al. (2016) proposed transferring mid-level representations learned from a labelled modality (RGB) to supervise training on a paired unlabeled modality (Depth). Tang et al. (2021) transferred knowledge from a multimodal teacher to an LM by pre-training the teacher on a video-text dataset and distilling knowledge to the student. Jin et al. (2022) explored enhancing LM abilities using a vision-language model for distilling vision knowledge into an LM using MLM (Devlin et al., 2018). Ghosh et al. (2023) utilized pre-trained LM to distil knowledge from language to vision. Initially trained on textual action sequences, the teacher model transfers its knowledge to a vision-based student for video-based action anticipation. Other notable contributions in transferring non-language modality information to language models include – VL-BERT (Su et al., 2019), VisualBERT (Li et al., 2019), ViLBERT (Lu et al., 2019) and X-adapter (Zhang et al., 2023), where multimodal models (pre-trained language models combined with visual encoders) are pre-trained on multimodal corpus and later evaluated on language understanding tasks.

**How is Our Method Different?** Unlike the existing cross-modal KD methods where only white-box teacher models are utilized after pre-training and/or fine-tuning, `ARMADA` can distil knowledge from any white or black-box vision-language model and that too without any pre-training. This ability widens its adaptability and improves the computational efficiency of the distillation process.

## 3 Proposed Methodology

In this section, we present `ARMADA` – an **A**lignment-induced c**r**oss-**mo**dal knowledge **d**istill**a**tion. We assume a multimodal model $F_t$ of modality set $\mathcal{M}_t$ as our teacher model. The student model $F_s$, the model that is being distilled, is assumed to belong to a different modality set $\mathcal{M}_s$, with $\mathcal{M}_s \subset \mathcal{M}_t$. We introduce `TS Aligner`, a cross-modal alignment module for aligning teacher and student models to a common modality. `TS Aligner`, denoted by $F_{ts}$, consists of a non-linear mapping $\tilde{F}_{ts}$, an output layer $O_{ts}$, a manifold projection layer $P_{ts}$ and an auxiliary output layer $O_{aux_{ts}}$. For a student input sequence $\{x_1, x_2, \cdots, x_n\} = X_s \in \mathcal{M}_s$, we assume a cross-modal representation $X_t \in \mathcal{M}_t$. For instance, if the student input is a sequence of text tokens, we can obtain the teacher sequence $X_t$ from a text-to-image teacher model. The cross-modal distillation process consists of three major steps – (i) output alignment, (ii) manifold alignment, and (iii) auxiliary output alignment. Figure 1 illustrates the overall framework of `ARMADA`. We enlist all the notations used in the paper and their descriptions in Table 1.

### 3.1 Output Alignment

For a given student input sequence $X_s$ and its equivalent teacher input sequence $X_t$, we first extract hidden representation using the student model $F_s$ and the frozen teacher model $F_t$, in sequence and obtain $h_s^X = F_s(X_s)$ and $h_t^X = F_t(X_t)$, respectively. We utilize a non-linear mapping, defined by the operators, $\tilde{F}_s$ and $\tilde{F}_{ts}$, for encoding the hidden representations, obtaining $h'^X_s = \tilde{F}_s(h_s^X)$ and $h'^X_{ts} = \tilde{F}_{ts}(h_t^X)$, respectively. For each task $\mathcal{T} \in \mathbb{T}$, we use task-specific linear layers for both `TS Aligner` and the student to obtain the outputs, $o_{ts}^{\mathcal{T}}$ and $o_s^{\mathcal{T}}$, respectively. We use the task-specific loss function $\mathcal{L}_{\mathcal{T}}$ (*e.g.,* cross-entropy for classification and mean squared error for regression) to compute loss against the ground-truth $Y^{\mathcal{T}}$ on the training dataset. For `TS Aligner`, the output loss is given as,

$$\mathcal{L}_{ts}^{(1)} = \mathcal{L}_{\mathcal{T}}(Y^{\mathcal{T}}, o_{ts}^{\mathcal{T}}). \tag{1}$$

| Notation Type | Notation | Description | Dimension/Shape |
|---|---|---|---|
| Modalities | $\mathcal{M}_t$ | Teacher model modality | |
| | $\mathcal{M}_s$ | Student model modality | |
| Architectures | $F_t$ | Multi-modal teacher model | |
| | $F_{ts}$ | TS Aligner module | |
| | $\tilde{F}_{ts}$ | Hidden non-linear projection mapping for TS Aligner | $\mathbb{R}^{128} \to \mathbb{R}^h$, $h$ is hidden dimension of TS Aligner |
| | $O_{ts}^{\mathcal{T}}$ | TS Aligner output layer for task $\mathcal{T}$ | $\mathbb{R}^h \to \mathbb{R}^c$, $h$ is hidden dimension of TS Aligner; $c$ is number of output classes for task $\mathcal{T}$ |
| | $P_{ts}$ | TS Aligner manifold projection mapping | $\mathbb{R}^h \to \mathbb{R}^{768}$, $h$ is hidden dimension of TS Aligner |
| | $O_{aux_{ts}}^{\mathcal{T}}$ | TS Aligner auxiliary output layer for task $\mathcal{T}$ | $\mathbb{R}^{768} \to \mathbb{R}^c$, $c$ is number of output classes for task $\mathcal{T}$ |
| | $F_s$ | Student representation learning model | |
| | $\tilde{F}_s$ | Hidden non-linear projection mapping for student model | $\mathbb{R}^d \to \mathbb{R}^d$, $d$ is hidden dimension of student language model |
| | $O_s^{\mathcal{T}}$ | Student output layer for task $\mathcal{T}$ | $\mathbb{R}^d \to \mathbb{R}^c$, $d$ is hidden dimension of student language model; $c$ is number of output classes for task $\mathcal{T}$ |
| | $P_s$ | Student manifold projection mapping | $\mathbb{R}^d \to \mathbb{R}^{768}$, $d$ is hidden dimension of student language model |
| | $O_{aux_s}^{\mathcal{T}}$ | Student auxiliary output layer for task $\mathcal{T}$ | $\mathbb{R}^{768} \to \mathbb{R}^c$, $c$ is number of output classes for task $\mathcal{T}$ |
| Inputs | $x_s$ | Sample student input sequence | |
| | $x_t$ | Teacher representation generated for $x_s$ | |
| | $X_s$ | Student input sequences $\{x_1, x_2, \cdots x_n\}$ | |
| | $X_t$ | Teacher generated sequences corresponding to $X_s$ | |
| | $Y^{\mathcal{T}}$ | Ground truth labels for inputs $X_s$ for task $\mathcal{T}$ | |
| Model outputs | $h_t^X$ | Representations obtained from multi-modal teacher model on input $X_t$ (inputs to TS Aligner) | $\mathbb{R}^{b \times 128}$, $b$ is batch size |
| | $h_{ts}^{'X}$ | TS Aligner hidden representation | $\mathbb{R}^{b \times h}$, $h$ is hidden dimension of TS Aligner; $b$ is batch size |
| | $o_{ts}^{\mathcal{T}}$ | TS Aligner output in task $\mathcal{T}$ | $\mathbb{R}^{b \times c}$, $c$ is number of output classes for task $\mathcal{T}$; $b$ is batch size |
| | $p_{ts}^X$ | TS Aligner manifold projection | $\mathbb{R}^{b \times 768}$, $b$ is batch size |
| | $o_{aux_{ts}}^{\mathcal{T}}$ | TS Aligner auxiliary output in task $\mathcal{T}$ | $\mathbb{R}^{b \times c}$, $c$ is number of output classes for task $\mathcal{T}$; $b$ is batch size |
| | $h_s^X$ | Hidden representation from student representation model | $\mathbb{R}^{b \times d}$, $d$ is hidden dimension of student language model; $b$ is batch size |
| | $h_s^{'X}$ | Student hidden representation | $\mathbb{R}^{b \times d}$, $d$ is hidden dimension of student language model; $b$ is batch size |
| | $o_s^{\mathcal{T}}$ | Student output in task $\mathcal{T}$ | $\mathbb{R}^{b \times c}$, $c$ is number of output classes for task $\mathcal{T}$; $b$ is batch size |
| | $p_s^X$ | Student manifold projection | $\mathbb{R}^{b \times 768}$, $b$ is batch size |
| | $o_{aux_s}^{\mathcal{T}}$ | Student auxiliary output in task $\mathcal{T}$ | $\mathbb{R}^{b \times c}$, $c$ is number of output classes for task $\mathcal{T}$; $b$ is batch size |
| Hyperparameters | $\alpha$ | Weightage of logit matching loss in student output loss | |
| | $\beta$ | Weightage of manifold alignment loss in the final loss value (for both TS Aligner and student models) | |
| | $\gamma$ | Weightage of auxiliary output loss in the final loss (for both TS Aligner and student models) | |
| | $\tau$ | Temperature value used in student logit matching loss | |

Table 1: Glossary of all the mathematical notations in the paper and their descriptions.

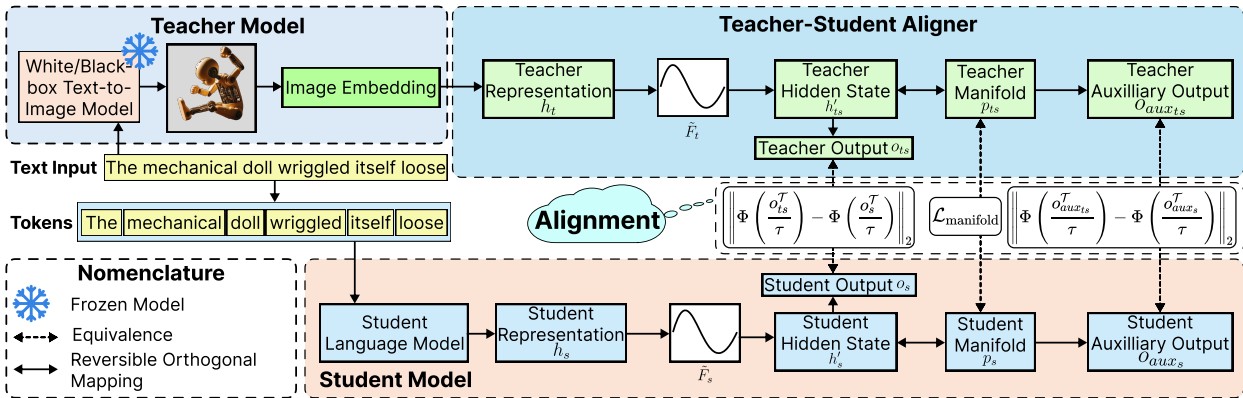

Figure 1: A schematic diagram of `ARMADA`. We highlight an example from the NLU task, where the text-to-image teacher model is used to distil knowledge to a language-only student model. Through manifold and output alignment loss objectives, the aligner model orchestrates the knowledge transfer from teacher to student modality.

For the student model, the output loss is computed as the weighted sum of the loss against the ground-truth and the logit matching loss (Hinton et al., 2015) with the aligner as,

$$\mathcal{L}_s^{(1)} = (1 - \alpha) \cdot \mathcal{L}_{\mathcal{T}}(Y^{\mathcal{T}}, o_s^{\mathcal{T}}) \tag{2}$$
$$+ \alpha \left\| \Phi\left(\frac{o_{ts}^{\mathcal{T}}}{\tau}\right) - \Phi\left(\frac{o_s^{\mathcal{T}}}{\tau}\right) \right\|_2.$$

For the classification task, $\Phi$ is the softmax function, whereas it is an identity function for the regression task. $\alpha$ and $\tau$ are the hyperparameters representing the output loss and temperature factors, respectively. We use $\alpha = 0.5$ as default.

### 3.2 Manifold Alignment

Sun et al. (2019) suggested an additional loss between the teacher and student hidden representations to enforce the student model in learning the feature representations similar to the teacher. However, minimizing the point-wise distance between the teacher and student representations can distort the information learned in the student's original modality in cross-modal distillation. Therefore, we first project the representations obtained from `TS Aligner` and the student models to a shared manifold and subsequently minimize their distance on the common subspace. We use two orthogonal projection mapping, $P_{ts}$ and $P_s$, to project the original teacher and student hidden representations to a common subspace to obtain $P_{ts}(h_{ts}^{'X}) = p_{ts}^X$ and $P_s(h_s^{'X}) = p_s^X$. Note that $p_{ts}^X$ and $p_s^X$ belong to the same dimensional subspace, therefore the manifold equivalence can be enforced with traditional similarity or distance measures. Following Huo et al. (2021), we propose an inner-product-based distance measure,

$$\mathcal{L}_{cosine}(p_{ts}^X, p_s^X) = 1 - \langle \frac{1}{n} \sum_{i=1}^n p_{ts}^{x_i}, \frac{1}{n} \sum_{i=1}^n p_s^{x_i} \rangle \tag{3}$$

where, $\langle ., . \rangle$ denotes the inner product between two vectors, and $\mathcal{L}_{cosine}$ measures the semantic similarity between teacher and student manifolds. Similarly, we formulate two other losses, $\mathcal{L}_{euclid}$ and $\mathcal{L}_{elementwise}$, to measure the distance between two manifolds in Euclidean space.

$$\mathcal{L}_{euclid}(p_{ts}^X, p_s^X) = \left\| \frac{1}{n} \sum_{i=1}^n p_{ts}^{x_i} - \frac{1}{n} \sum_{i=1}^n p_s^{x_i} \right\|_2 \tag{4}$$

$$\mathcal{L}_{elementwise}(p_{ts}^X, p_s^X) = \frac{1}{n} \sum_{i=1}^n ||p_{ts}^{x_i} - p_s^{x_i}||_2 \tag{5}$$

Using $\mathcal{L}_{euclid}$, we compute Euclidean distance between the centroid of the teacher and student projection vectors, whereas, with $\mathcal{L}_{elementwise}$, we compute the expected pairwise (element-wise) distances between the teacher and student projection vectors. Different loss formulations induce different student model regularizations, encouraging the student to learn different cross-modal abstractions. For the rest of the paper, we generalize these three losses as $\mathcal{L}_{manifold}$.

**Proposition 1.** $\mathcal{L}_{elementwise}$ enforces highest regularization effect than $\mathcal{L}_{euclid}$ and $\mathcal{L}_{cosine}$.

**Proof.** We would like to show that $\mathcal{L}_{elementwise}(p_{ts}^X, p_s^X) \geq \mathcal{L}_{euclid}(p_{ts}^X, p_s^X) > ||\overline{p_{ts}^X}|| \cdot ||\overline{p_s^X}|| \cdot \sqrt{\mathcal{L}_{cosine}(p_{ts}^X, p_s^X)}$. From Equation 4 we obtain,

$$\mathcal{L}_{euclid}(p_{ts}^X, p_s^X) = \left\| \frac{1}{n} \sum_{i=1}^{n} (p_{ts}^{x_i} - p_s^{x_i}) \right\|_2.$$

$$\leq \frac{1}{n} \sum_{i=1}^{n} \left\| p_{ts}^{x_i} - p_s^{x_i} \right\|_2 \text{(Triangle inequality)}$$

$$= \mathcal{L}_{elementwise}(p_{ts}^X, p_s^X)$$

We denote $\frac{1}{n} \sum_{i=1}^{n} p_{ts}^{x_i}$ as $\overline{p_{ts}^X}$ and $\frac{1}{n} \sum_{i=1}^{n} p_s^{x_i}$ as $\overline{p_s^X}$. Then from Equations 3 and 4, we obtain,

$$\mathcal{L}_{euclid}^2(p_{ts}^X, p_s^X) = \left\| \overline{p_{ts}^X} - \overline{p_s^X} \right\|_2^2.$$

$$= \left\| \overline{p_{ts}^X} \right\|_2^2 + \left\| \overline{p_s^X} \right\|_2^2 - 2 \left\| \overline{p_{ts}^X} \right\|_2 \left\| \overline{p_s^X} \right\|_2 < \overline{p_{ts}^X}, \overline{p_s^X} >$$

From AM-GM inequality

$$\geq 2 \left\| \overline{p_{ts}^X} \right\|_2 \left\| \overline{p_s^X} \right\|_2 - 2 \left\| \overline{p_{ts}^X} \right\|_2 \left\| \overline{p_s^X} \right\|_2 < \overline{p_{ts}^X}, \overline{p_s^X} >$$

$$= 2 \left\| \overline{p_{ts}^X} \right\|_2 \left\| \overline{p_s^X} \right\|_2 \mathcal{L}_{cosine}(p_{ts}^X, p_s^X)$$

$$> \left\| \overline{p_{ts}^X} \right\|_2 \left\| \overline{p_s^X} \right\|_2 \mathcal{L}_{cosine}(p_{ts}^X, p_s^X).$$

### 3.3 Auxiliary Output Alignment

To further regularize the student representations, we incorporate an auxiliary output head on `TS Aligner` and student projection heads. We align the models to learn appropriate manifold projections that maximize their performance on downstream tasks by incorporating an auxiliary output head on both teacher and student projection vectors. Given the auxiliary output $o_{aux_{ts}}^{\mathcal{T}} = O_{aux_{ts}}^{\mathcal{T}}(p_{ts}^X)$ and $o_{aux_s}^{\mathcal{T}} = O_{aux_s}^{\mathcal{T}}(p_s^X)$, we calculate the auxiliary losses, $\mathcal{L}_{ts}^{(2)}$ and $\mathcal{L}_s^{(2)}$, similar to Equations 1 and 2. We establish homeomorphism between topological spaces to understand how the intertwining of different alignments ensure the effectiveness of multimodal knowledge transfer.

**Definition 1.** A function $f : X \rightarrow Y$ is called *homeomorphism* if $f$ is bijective (*i.e.,* one-to-one and onto) and continuous, and the inverse function $f^{-1}$ exists and is continuous too.

**Definition 2.** Two topological spaces, $X$ and $Y$, are called *homeomorphic* if a homeomorphism exists between them. Homeomorphism ensures that two spaces are topologically similar and exhibit structural equivalence.

**Proposition 2.** Let's assume a sequence of inputs $X$ and their corresponding aligner projection space $p_{ts}^X$, student projection space $p_s^X$, teacher output space $o_{ts}^X$ and student output space $o_s^X$, auxiliary teacher output space $o_{aux_{ts}}^X$ and auxiliary student output space $o_{aux_s}^X$. If $p_{ts}^X$ and $p_s^X$ are homeomorphic, $o_{ts}^X$ and $o_s^X$ are also homeomorphic. Similarly, $o_{aux_{ts}}^X$ and $o_{aux_s}^X$ are also homeomorphic.

**Proof.** If $p_{ts}^X$ and $p_s^X$ are homeomorphic, then there exists a continuous bijective function $f : p_{ts}^X \rightarrow p_s^X$, so that the inverse $f^{-1} : p_s^X \rightarrow p_{ts}^X$ is also continuous. Figure 2 highlights the mapping between different spaces.

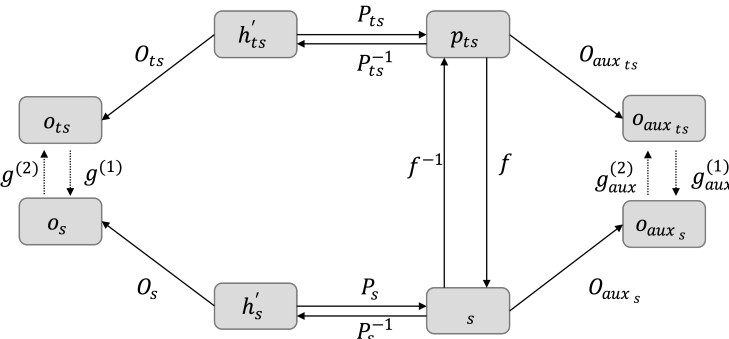

Figure 2: Commutative diagram between the manifold, output and auxiliary output spaces. The bold lines denote the functions already defined between the spaces. The dashed lines indicate the functions we want to establish in the proof of Proposition 2.

Being an orthogonal projection matrix, $P_{ts}$ has its inverse $P_{ts}^T$. Similarly, the inverse of $P_s$ exists. Being linear maps, all the functions $P_{ts}, P_s, O_{ts}, O_s, O_{aux_{ts}}$ and $O_{aux_s}$ are continuous. To prove that the output spaces $o_{ts}^X$ and $o_s^X$ are homeomorphic, we need to establish two continuous bijection functions $g^{(1)}: o_{ts}^X \to o_s^X$ and $g^{(2)}: o_s^X \to o_{ts}^X$, such that $g^{(1)} \circ g^{(2)} = g^{(2)} \circ g^{(1)} = I$. Being finite spaces with the same cardinality (same as the batch size), the surjectivity of functions $P_{ts}, P_s, O_{ts}, O_s, O_{aux_{ts}}$ and $O_{aux_s}$, suggests injectivity of these functions. Therefore, all these functions are bijections.

$$g^{(1)} \circ O_{ts} \circ P_{ts}^{-1} = O_s \circ P_s^{-1} \circ f. \tag{6}$$

Similarly, for a given point $p_s^x \in P_s$,

$$g^{(2)} \circ O_s \circ P_s^{-1} = O_{ts} \circ P_{ts}^{-1} \circ f^{-1}$$
$$\implies g^{(1)} \circ g^{(2)} \circ O_s \circ P_s^{-1} = g^{(1)} \circ O_{ts} \circ P_{ts}^{-1} \circ f^{-1}$$
$$\implies g^{(1)} \circ g^{(2)} \circ O_s \circ P_s^{-1} = O_s \circ P_s^{-1} \circ f \circ f^{-1} (\text{Eq } 6)$$
$$\implies g^{(1)} \circ g^{(2)} \circ O_s \circ P_s^{-1} = O_s \circ P_s^{-1}.$$

The function $O_s$ has a right inverse due to its surjectivity. Similarly, $O_{ts}$ also has a right inverse. Therefore, we get $g^{(1)} \circ g^{(2)} = I$. A similar formulation leads us to $g^{(2)} \circ g^{(1)} = I$. Additionally, we can simplify

$$g^{(1)} = O_s \circ P_s^{-1} \circ f \circ P_{ts} \circ O_{ts}^{-1}. \tag{7}$$

As a composition of continuous bijections, $g^{(1)}$ is also a continuous bijection. Therefore, $g^{(1)}$ is a homeomorphism. A similar formulation can be used to establish that $g_{aux}^{(1)}$ is a homeomorphism. Therefore, a homeomorphic relationship between $p_{ts}^X$ and $p_s^X$ ensures homeomorphic relationship between $o_{ts}^X$ and $o_s^X$ and homeomorphic relationship between $o_{aux_{ts}}^X$ and $o_{aux_s}^X$.

Proposition 2 suggests how an equivalence between the teacher and student manifolds can establish equivalence between their associated output and auxiliary output spaces. Therefore, we can regularize the student by minimizing the distance from the teacher manifold. As opposed to vokenization (Tan & Bansal, 2020) and VidLanKD (Tang et al., 2021), which use pre-training objectives to train the teacher models, we optimize `TS Aligner` on task-specific objectives, allowing task-specific knowledge diffusion to students. The final `TS Aligner` loss is calculated as $\mathcal{L}_t = \mathcal{L}_{ts}^{(1)} + \gamma \mathcal{L}_t^{(2)}$. For the student model, we use the output loss, auxiliary output

loss, and manifold alignment loss to calculate the final training loss as $\mathcal{L}_s = \mathcal{L}_s^{(1)} + \gamma\mathcal{L}_s^{(2)} + \beta\mathcal{L}_{manifold}(p_t^X, p_s^X)$. $\beta$ and $\gamma$ are hyperparameters used to assign weightage to the manifold and auxiliary output alignment loss, respectively. We use $\beta = 1$ and $\gamma = 1$ by default.

## 4 Experimental Setup

**Tasks.** We apply `ARMADA` on seven student LMs and four multimodal teacher models and compare it against ten unimodal and three multimodal KD methods across 12 NLU tasks and 5 instruction-following tasks. Following Liu et al. (2022); Zhou et al. (2021), we consider four different NLU tasks from SuperGLUE (Wang et al., 2019) – CB, COPA, BoolQ and WiC, and eight tasks from GLUE (Wang et al., 2018) – CoLA, MRPC, RTE, STS-B (categorized as *GLUE-small*), SST-2, MNLI, QNLI and QQP (categorized as *GLUE-large*). We also consider Multimodal-IMDb (Arevalo et al., 2017), a classification task to predict movie genres based on poster images (image modality) and additional metadata, including plot summaries (text modality), etc. Additionally, we consider five commonsense reasoning tasks – HellaSwag (Zellers et al., 2019), Wino-Grande (Sakaguchi et al., 2021), ARC-easy, ARC-challenge (Clark et al., 2018) and OpenBookQA (Mihaylov et al., 2018) and three mathematical reasoning tasks – MultiArith (Roy & Roth, 2016), SingleEq (Koncel-Kedziorski et al., 2016) and AQuA (Ling et al., 2017). For instruction-following experiments, we fine-tune the student models on Dolly-15K Gu et al. (2024) dataset and evaluate on Dolly, SelfInst Wang et al. (2022a), Vicuna Chiang et al. (2023), SNI Wang et al. (2022b) and UNI Honovich et al. (2023) datasets.

**Student and teacher models.** We consider seven different pre-trained language models, including BERT-base (Devlin et al., 2018) ($111M$ parameters), pre-trained BERT 6-layer ($66M$ parameters), DeBERTa-v2-xxlarge (He et al., 2020) ($1.4B$ parameters), OPT-1.3B (Zhang et al., 2022), LLaMA-7B (Touvron et al., 2023), LLaMA-3.1-8B (Dubey et al., 2024) and LLaMA-3.2-3B*. We utilize two text-to-image teacher models: Stable Diffusion v1-4 (Rombach et al., 2022a) and Midjourney (Borji, 2022), along with the ModelScope text-to-video (Wang et al., 2023) ($1.7B$ parameters) and the SpeechT5 text-to-audio model (Ao et al., 2022) ($144M$ parameters). For the text-to-image teacher models, we generate latent representations that are $32 \times 32$ dimensions (flattened to a 1024-dimensional vector) for each text input, using 20 diffusion inference steps and a guidance scale value of 7.5. Similar configurations apply for extracting latent representations from the text-to-video model, where 64 frames are generated for each text input. We extract the hidden representation from the first frame, resulting in a $512 \times 512$ hidden representation that is flattened into a 262144-dimensional vector for each text input. For the text-to-audio teacher model, we generate audio corresponding to each text input, which is subsequently processed by the Wav2Vec2-base model (Baevski et al., 2020) to produce a 768-dimensional vector representation. These vector representations of the texts are utilized to pre-train the `TS Aligner`. For natural language understanding (NLU) tasks, we employ cross-entropy loss for the pre-training of the `TS Aligner`. Conversely, for commonsense reasoning and instruction-tuning tasks, the `TS Aligner` is pre-trained using reconstruction loss measured by the mean squared error.

**Evaluation metrics.** For CoLA and STS-B tasks, we use the Mathews correlation coefficient ('Mcc') and Pearson's correlation coefficient ('Pear'), respectively, for evaluation. For the remaining NLU and commonsense-reasoning tasks, we use the accuracy score ('Acc'). For evaluating on instruction-tuning tasks, we use 'RougeL' metric.

**Baselines.** The competing baselines include unimodal KD methods (KD (Hinton et al., 2015), Sequence-KD (Kim & Rush, 2016), PKD (Sun et al., 2019), TinyBERT (Jiao et al., 2019), MetaDistil (Zhou et al., 2021) and Rail KD (Haidar et al., 2022)) and multimodal KD methods (VL-BERT (Su et al., 2019), VisualBERT (Li et al., 2019), ViLBERT (Lu et al., 2019), Vokenization (Tan & Bansal, 2020), VidLanKD (Tang et al., 2021), Visual guidance (Zhang et al., 2021) and X-adapter (Zhang et al., 2023)). Appendix B presents more details.

---

*The pre-trained models are obtained from `https://huggingface.co/models/`.

| Models | Distillation type | CoLA Mcc | MRPC Acc | RTE Acc | STS-B Pear | CB Acc | COPA Acc | BoolQ Acc | WiC Acc |
|---|---|---|---|---|---|---|---|---|---|
| BERT-6L | undistilled | 42.8±0.1 | 78.6±0.0 | 64.1±0.2 | 88.5±0.4 | 75.0±0.0 | 53.0±0.0 | 71.3±0.1 | 55.7±0.2 |
| | ARMADA - $\mathcal{L}_{cosine}$ | **46.1±0.1** | **80.3±0.0** | **67.5±0.0** | 88.5±0.1 | **76.8±0.0** | **62.8±0.1** | **73.1±0.0** | **57.8±0.1** |
| | ARMADA - $\mathcal{L}_{euclid}$ | **47.6±0.0** | **80.5±0.0** | **67.4±0.1** | 88.5±0.2 | **78.5±0.1** | **63.0±0.0** | **73.1±0.0** | **58.1±0.1** |
| | ARMADA - $\mathcal{L}_{elementwise}$ | **45.7±0.2** | **79.5±0.0** | **67.5±0.1** | 88.6±0.1 | **78.6±0.0** | **63.0±0.0** | **73.0±0.1** | **57.9±0.0** |
| BERT-base | undistilled | 60.8±0.1 | 83.8±0.1 | 71.0±0.0 | 89.3±0.0 | 83.7±0.1 | 62.9±0.0 | 75.1±0.1 | 58.4±0.1 |
| | ARMADA - $\mathcal{L}_{cosine}$ | **61.6±0.0** | **85.0±0.0** | **72.5±0.1** | **89.5±0.1** | **85.7±0.0** | **67.7±0.2** | **75.8±0.0** | 58.5±0.0 |
| | ARMADA - $\mathcal{L}_{euclid}$ | **61.9±0.0** | **85.2±0.1** | **73.9±0.0** | **89.7±0.0** | **85.7±0.0** | **67.9±0.1** | 75.1±0.0 | **58.8±0.0** |
| | ARMADA - $\mathcal{L}_{elementwise}$ | **61.8±0.0** | **85.0±0.0** | **71.7±0.1** | **89.6±0.0** | **87.4±0.1** | **68.0±0.0** | **75.3±0.0** | 58.4±0.0 |
| DeBERTa-v2-xxlarge | undistilled | 72.1±0.0 | 89.7±0.1 | 90.3±0.0 | 92.3±0.0 | 81.9±0.1 | 83.7±0.0 | 87.2±0.0 | 60.8±0.1 |
| | ARMADA - $\mathcal{L}_{cosine}$ | **72.6±0.0** | 89.9±0.0 | **92.3±0.0** | **92.4±0.1** | **92.8±0.0** | 81.9±0.1 | 86.2±0.0 | 59.2±0.0 |
| | ARMADA - $\mathcal{L}_{euclid}$ | **73.1±0.0** | **90.2±0.0** | **91.6±0.1** | 92.3±0.0 | **92.8±0.0** | 78.9±0.1 | **87.7±0.0** | 58.2±0.0 |
| | ARMADA - $\mathcal{L}_{elementwise}$ | **73.4±0.0** | 89.4±0.1 | **91.6±0.0** | 92.0±0.1 | **93.0±0.0** | 79.9±0.1 | **87.8±0.0** | 59.9±0.0 |
| OPT-1.3B | undistilled | 59.6±0.0 | 84.3±0.1 | 76.3±0.0 | 89.7±0.0 | 78.2±0.1 | 54.8±0.0 | 74.6±0.0 | 57.9±0.1 |
| | ARMADA - $\mathcal{L}_{cosine}$ | **62.6±0.0** | 84.5±0.0 | **77.3±0.0** | **90.4±0.1** | **78.5±0.0** | **57.9±0.1** | **77.3±0.0** | 56.7±0.0 |
| | ARMADA - $\mathcal{L}_{euclid}$ | **61.7±0.0** | 83.1±0.1 | **79.0±0.0** | **90.1±0.0** | 76.7±0.1 | **57.0±0.0** | **76.0±0.0** | 57.0±0.0 |
| | ARMADA - $\mathcal{L}_{elementwise}$ | **62.4±0.0** | **84.7±0.0** | **78.6±0.1** | **90.0±0.0** | 76.7±0.1 | **58.9±0.0** | **76.1±0.0** | 56.1±0.0 |

Table 2: Results of different LMs (undistilled and distilled by ARMADA with Stable Diffusion teacher) on **GLUE-small** and **SuperGLUE** tasks - different training paradigms. **Blue** indicates the tasks where the student model distilled - ARMADA exhibits better performance than the undistilled fine-tuned variants. We report the average (and standard deviation) scores from three runs for each experiment.

| Models | Distillation type | MNLI Acc | QNLI Acc | QQP Acc | SST-2 Acc |
|---|---|---|---|---|---|
| BERT-6L | undistilled | 77.7±0.0 | 84.9±0.1 | 89.0±0.0 | 87.6±0.1 |
| | ARMADA - $\mathcal{L}_{cosine}$ | **80.2±0.0** | **87.3±0.0** | 88.6±0.1 | **90.7±0.0** |
| | ARMADA - $\mathcal{L}_{euclid}$ | **80.3±0.0** | **87.2±0.1** | 88.9±0.0 | **90.6±0.0** |
| | ARMADA - $\mathcal{L}_{elementwise}$ | **80.1±0.0** | **87.3±0.0** | **89.1±0.1** | **90.7±0.0** |
| BERT-base | undistilled | 79.3±0.0 | 87.8±0.1 | 83.1±0.0 | 89.2±0.1 |
| | ARMADA - $\mathcal{L}_{cosine}$ | **83.4±0.0** | **91.0±0.0** | **89.4±0.1** | **93.1±0.0** |
| | ARMADA - $\mathcal{L}_{euclid}$ | **83.5±0.0** | **91.0±0.0** | **88.8±0.1** | **92.0±0.0** |
| | ARMADA - $\mathcal{L}_{elementwise}$ | **83.4±0.0** | **91.2±0.0** | **88.9±0.1** | **92.8±0.0** |

Table 3: Results of BERT models (undistilled and distilled by ARMADA with Stable Diffusion teacher) on **GLUE-large** tasks. **Blue** indicates cases where ARMADA improves performance of the undistilled model.

## 5 Experimental Results

### 5.1 Results on NLU and Reasoning Tasks

**Effectiveness of ARMADA on different LLMs.** Table 2 shows the performance improvement of student LMs after ARMADA-based distillation compared to their undistilled counterparts on GLUE-small and SuperGLUE benchmark tasks. Table 3 reports the performance of BERT models on GLUE-large tasks. Our findings reveal a consistent and significant enhancement in model capabilities for different pre-trained models with varied complexities. Once distilled using ARMADA, BERT-base and BERT-6L models demonstrate average improvements of 2.8% and 3.4%, respectively, across all tasks (combining Tables 2 and 3), outperforming their undistilled counterparts. A similar trend is also observed in larger LMs, with DeBERTa and OPT recording improvements of 1.4% and 1.5%, respectively. Table 4 highlights the results for LLaMA-7B, demonstrating an average improvement of 0.5%. Even in the zero-shot regime, ARMADA boosts the performance of a fine-tuned LLaMA-7B model on five out of eight reasoning tasks with a maximum improvement margin of 2.6%. One-sided t-tests suggest that the performance improvement over the undistilled model is statistically significant (for GLUE-small tasks $t$-statistic $= 3.98$ with $p$-value $= 0.00$, for GLUE-large $t$-statistic $= 5.11$ with $p$-value $= 0.00$ and for reasoning tasks $t$-statistic $= 2.0$ with $p$-value $= 0.04$), indicating the effectiveness of cross-modal distillation on language understanding and generation tasks. These results are particularly remarkable as the cross-modal teacher representation learning (stable diffusion model in this case) is not fine-tuned on the language tasks, underscoring the robust latent linguistic structure present in vision-language representations.

**ARMADA with different teacher.** To highlight the effectiveness of ARMADA under a different text-to-image teacher model, we highlight the results obtained by different LMs distilled with ARMADA with Midjourney (Borji,

| Distillation type | Hellaswag | Winogrande | ARC-c | ARC-e | OpenBookQA | MultiArith | SingleEq | AQuA | Avg. |
|---|---|---|---|---|---|---|---|---|---|
| undistilled | 65.2 | 70.5 | 65.5 | 80.9 | 75.6 | 96.0 | 83.5 | 14.8 | 69.0 |
| ARMADA with $\mathcal{L}_{cosine}$ | **67.8** | 69.5 | 64.7 | **81.1** | **76.8** | **97.0** | 83.1 | **15.9** | **69.5** |
| ARMADA with $\mathcal{L}_{euclid}$ | **67.0** | 70.0 | **65.7** | 80.8 | **76.5** | **96.5** | 83.3 | **15.6** | **69.4** |
| ARMADA with $\mathcal{L}_{elementwise}$ | **65.8** | 69.6 | 64.3 | 80.8 | **75.9** | **96.2** | 83.1 | **15.2** | 68.9 |

Table 4: Zero-shot accuracy on reasoning tasks for LLaMA-7B distilled with `ARMADA` with Stable Diffusion teacher model. **Blue** indicates the tasks where `ARMADA` improves performance over the undistilled LLaMA.

| Models | Distillation type | CoLA | MRPC | RTE | STS-B | CB | COPA | BoolQ | WiC |
|---|---|---|---|---|---|---|---|---|---|
| | | Mcc | Acc | Acc | Pear | Acc | Acc | Acc | Acc |
| BERT-6L | undistilled | 42.8±0.1 | 78.6±0.0 | 64.1±0.2 | 88.5±0.4 | 75.0±0.0 | 53.0±0.0 | 71.3±0.1 | 55.7±0.2 |
| | ARMADA - $\mathcal{L}_{cosine}$ | **44.0±0.0** | **80.1±0.0** | **65.2±0.1** | 87.9±0.0 | 73.1±0.1 | 49.9±0.0 | **71.8±0.0** | **56.6±0.0** |
| | ARMADA - $\mathcal{L}_{euclid}$ | **42.9±0.0** | **79.1±0.1** | **66.3±0.0** | 88.3±0.0 | **75.0±0.0** | **62.9±0.1** | **72.4±0.0** | **58.2±0.0** |
| | ARMADA - $\mathcal{L}_{elementwise}$ | **46.9±0.0** | **79.3±0.0** | **67.0±0.1** | 88.3±0.0 | **78.4±0.1** | 51.9±0.0 | **73.3±0.0** | 55.9±0.0 |
| BERT-base | undistilled | 60.8±0.1 | 83.8±0.1 | 71.0±0.0 | 89.3±0.0 | 83.7±0.1 | 62.9±0.0 | 75.1±0.1 | 58.4±0.1 |
| | ARMADA - $\mathcal{L}_{cosine}$ | 58.8±0.0 | **84.9±0.0** | 69.9±0.1 | 88.6±0.0 | **85.6±0.1** | **64.9±0.0** | 73.3±0.0 | 56.2±0.0 |
| | ARMADA - $\mathcal{L}_{euclid}$ | 60.1±0.0 | **84.1±0.1** | 68.8±0.0 | 88.3±0.0 | **87.4±0.1** | 62.9±0.0 | 72.3±0.0 | 58.2±0.0 |
| | ARMADA - $\mathcal{L}_{elementwise}$ | **61.1±0.0** | **85.0±0.0** | 68.8±0.1 | 88.3±0.0 | **83.8±0.1** | 60.9±0.0 | 73.0±0.0 | **58.7±0.0** |
| DeBERTa-v2-xxlarge | undistilled | 72.1±0.0 | 89.7±0.1 | 90.3±0.0 | 92.3±0.0 | 81.9±0.1 | 83.7±0.0 | 87.2±0.0 | 60.8±0.1 |
| | ARMADA - $\mathcal{L}_{cosine}$ | 71.4±0.0 | **90.3±0.0** | **90.9±0.1** | 91.9±0.0 | **94.5±0.1** | 81.9±0.0 | **87.3±0.0** | 57.6±0.0 |
| | ARMADA - $\mathcal{L}_{euclid}$ | 68.2±0.0 | **91.1±0.1** | **90.8±0.0** | 91.8±0.0 | **93.4±0.1** | 79.9±0.0 | **87.9±0.0** | 60.0±0.0 |
| | ARMADA - $\mathcal{L}_{elementwise}$ | 70.4±0.0 | **90.5±0.0** | **90.3±0.1** | 92.1±0.0 | **92.8±0.1** | 81.9±0.0 | **88.1±0.0** | 59.6±0.0 |
| OPT-1.3B | undistilled | 59.6±0.0 | 84.3±0.1 | 76.3±0.0 | 89.7±0.0 | 78.2±0.1 | 54.8±0.0 | 74.6±0.0 | 57.9±0.1 |
| | ARMADA - $\mathcal{L}_{cosine}$ | **60.4±0.0** | 83.4±0.0 | 74.2±0.1 | **90.0±0.1** | **80.2±0.0** | 54.9±0.0 | **75.3±0.0** | 55.9±0.0 |
| | ARMADA - $\mathcal{L}_{euclid}$ | **61.2±0.0** | 82.4±0.1 | **77.1±0.0** | 89.9±0.0 | **78.5±0.1** | **58.9±0.0** | **75.6±0.0** | 57.0±0.0 |
| | ARMADA - $\mathcal{L}_{elementwise}$ | 59.5±0.0 | 80.5±0.1 | **77.1±0.0** | **90.1±0.1** | 76.6±0.0 | **58.9±0.0** | **76.0±0.0** | 57.0±0.0 |

Table 5: Results of different LMs (undistilled and distilled by `ARMADA`) on GLUE-small and SuperGLUE tasks with different training paradigms with Midjourney (Borji, 2022) teacher model.

2022) teacher in Table 5. Midjourney model uses Stable Diffusion model with LoRA (Hu et al., 2021) adapter on $100K+$ midjourney images. As highlighted in Table 6, even with Midjourney teacher model, `TS Aligner` is significantly inferior to the student models on downstream tasks. Despite that, the aligner model effectively imparts cross-modal knowledge to the student model. BERT-base and BERT-6L models distilled with `ARMADA` achieve an average improvement of 0.44% and 3.2%, respectively, over the undistilled counterparts.

| Teacher Model | CoLA | MRPC | RTE | STS-B | CB | COPA | BoolQ | WiC |
|---|---|---|---|---|---|---|---|---|
| | Mcc | Acc | Acc | Pear | Acc | Acc | Acc | Acc |
| Stable Diffusion | 7.8 | 66.6 | 53.4 | 3.7 | 55.4 | 62.0 | 60.9 | 53.0 |
| Midjourney | 1.8 | 66.5 | 52.3 | 1.9 | 50.0 | 56.0 | 56.6 | 50.6 |

Table 6: Results of `TS Aligner` on all GLUE-small and SuperGLUE tasks with different teacher models.

**Comparison against baselines.** We further compare the improvement shown by `ARMADA` against the contemporary unimodal and multimodal KD baselines in Tables 7 and 8, respectively.[†] On GLUE tasks, `ARMADA` provides the best performance boost for the BERT-6L student model, with an average improvement of 2.2%, even higher than the most competitive unimodal KD baseline MetaDistil which uses $100\times$ more trainable teacher parameters for knowledge distillation. On GLUE-large tasks, `ARMADA` exhibits comparable performance among all the multimodal distillation methods. Strikingly, even with significantly fewer teacher training steps than the most competitive multimodal baselines VidLanKD and X-adapter ($< 0.8\%$), `ARMADA` provides the competitive performance for the BERT-base student model. One-tailed t-statistic value of 1.56 ($p$-value $< 0.1$) highlights the statistical significance of the improvements shown by `ARMADA` over the unimodal and multimodal KD baselines.

**Results on multimodal task.** We highlight the performance obtained by BERT-12L on the MM-IMDb classification task in Table 9. With cross-modal distillation, the BERT-12L model achieves a $+1.2\%$ improvement in macro-F1 and $+1.3\%$ improvement in micro-F1 over the undistilled baseline. Notably, this gain is comparable to the improvement reported by Kiela et al. (2019) using a fully multimodal BERT architecture. Importantly, in our setting the teacher and student share no common modality at inference

[†]Following Zhang et al. (2021), we only compare the multimodal KD methods on GLUE-large tasks.

| Methods | CoLA | MNLI | MRPC | QNLI | QQP | RTE | SST-2 | STS-B | Avg. |
|---|---|---|---|---|---|---|---|---|---|
| KD (Hinton et al., 2015) | 43.5 | 79.3 | 79.4 | 85.6 | 89.6 | 64.4 | 87.8 | 89.2 | 77.4 |
| PKD (Sun et al., 2019) | 43.9 | 79.4 | 78.9 | 85.9 | 89.6 | 64.3 | 87.9 | 89.2 | 77.4 |
| TinyBERT (Jiao et al., 2019) | 41.8 | **80.3** | 80.7 | 86.2 | 89.3 | 64.4 | 88.5 | 89.8 | 77.6 |
| RCO (Jin et al., 2019) | 43.0 | 79.1 | 79.3 | 86.1 | 89.3 | 64.3 | 88.0 | 89.2 | 77.3 |
| TAKD (Mirzadeh et al., 2020) | 43.2 | 79.2 | 79.2 | 86.0 | 89.4 | 65.0 | 88.0 | 89.3 | 77.4 |
| DML (Zhang et al., 2018) | 43.1 | 79.1 | 79.3 | 86.0 | 89.0 | 64.9 | 88.1 | 89.2 | 77.3 |
| ProKT (Shi et al., 2020) | 43.7 | 79.5 | 80.5 | 86.1 | 89.6 | 64.9 | 87.9 | 89.6 | 77.7 |
| SFTN (Park et al., 2021) | 43.0 | 79.1 | 79.5 | 85.9 | 89.1 | 65.0 | 88.1 | 89.3 | 77.4 |
| MetaDistil (Zhou et al., 2021) | **48.0** | 80.2 | 81.0 | 86.8 | 89.7 | 66.1 | 88.9 | **90.7** | 78.9 |
| Continuation KD (Jafari et al., 2022) | 42.7 | 78.6 | **84.3** | 86.0 | **91.3** | 64.9 | 89.7 | 89.1 | 78.3 |
| Rail KD (Haidar et al., 2022) | 45.9 | 78.6 | 79.3 | 85.7 | 90.0 | **68.3** | 89.4 | 90.0 | 78.4 |
| ARMADA | 47.6 | **80.3** | 80.5 | **87.3** | 89.2 | 67.5 | **90.7** | 88.7 | **79.0** |

Table 7: Improvement for BERT-6L on GLUE tasks distilled with ARMADA and other contemporary unimodal distillation methods. For all the KD baselines except ARMADA, BERT-base (110*M* learnable parameters) is used as the teacher model. ARMADA uses Stable Diffusion as the teacher model (0.8*M* learnable parameters).

| Methods | SST-2 | QNLI | QQP | MNLI | Avg. |
|---|---|---|---|---|---|
| ViLBERT | 90.3 | 89.6 | 88.4 | 82.4 | 87.7 |
| VL-BERT | 90.1 | 89.5 | 88.6 | 82.9 | 87.8 |
| VisualBERT | 90.3 | 88.9 | 88.4 | 82.4 | 87.5 |
| Img-Voken | 92.2 | 88.6 | 88.6 | 82.6 | 88.0 |
| Visual Guidance | 89.7 | 88.2 | 84.2 | 79.5 | 85.4 |
| Vid-Voken w/ResNet | 93.4 | 89.2 | 88.7 | 83.0 | 88.6 |
| Vid-Voken w/CLIP | **94.1** | 89.8 | 89.0 | 83.9 | 89.2 |
| X-adapter | 92.7 | **91.4** | 89.3 | **84.4** | **89.4** |
| ARMADA | 93.1 | 91.3 | **89.5** | 83.5 | **89.4** |

Table 8: Improvement for BERT-base model on GLUE-large tasks distilled by ARMADA with Stable Diffusion teacher model, compared against the improvements reported by multimodal KD methods.

time, demonstrating that ARMADA effectively transfers multimodal priors into a unimodal language model without architectural modification.

To further validate the generality of this effect, we additionally evaluate on the Hateful Memes benchmark (Kiela et al., 2020), a more challenging multimodal dataset designed to prevent unimodal shortcuts. The undistilled BERT-12L achieves 55.4% accuracy, whereas ARMADA improves performance consistently across alignment variants to 60.2–60.4% accuracy, corresponding to an absolute gain of approximately 5%. Unlike MM-IMDb, where improvements are modest but consistent, Hateful Memes exhibits a substantially larger gain, suggesting that cross-modal distillation is particularly beneficial for tasks requiring joint reasoning over visual and textual semantics.

Crucially, in both experiments, the student remains a text-only model at inference time, underscoring that the observed gains arise from transferred cross-modal semantic structure rather than architectural multimodality.

| Models | Distillation type | MM-IMDb F1 Micro/Macro | Meme Acc |
|---|---|---|---|
| | undistilled | 64.9 48.1 | 55.4 |
| BERT-12L | ARMADA - $\mathcal{L}_{cosine}$ | **65.7/ 49.3** | **60.2** |
| | ARMADA - $\mathcal{L}_{euclid}$ | **66.2/ 48.9** | **60.4** |
| | ARMADA - $\mathcal{L}_{elementwise}$ | **65.3/ 48.0** | **59.4** |

Table 9: Results of BERT-base model - and without cross-modal distillation on MM-IMDb genre and Hateful meme classification tasks.

**Auxiliary output performance.** We report the performance obtained with the auxiliary heads in Table 10. The results reveal a striking Spearman rank correlation between output head and auxiliary head performances of 0.99 for student models and 0.69 for teacher models. This correlation indicates a strong alignment in performance trends despite the auxiliary output head's generally lower performance, attributable to its operation over presumed lower-dimensional manifold projections. On average, student models exhibit a modest improvement of 0.61% with the output head over the auxiliary

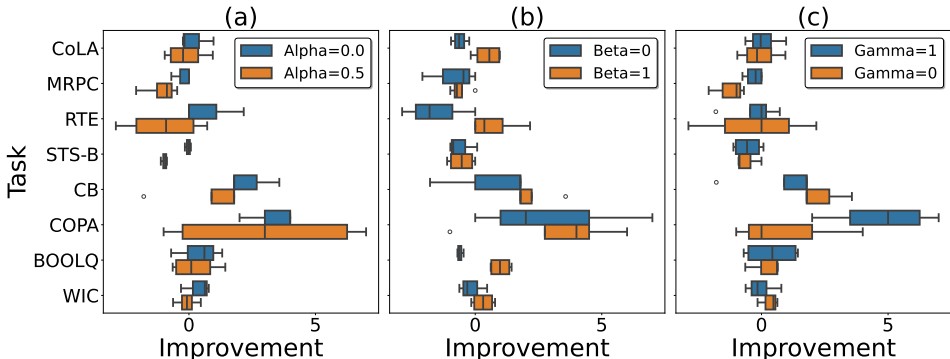

Figure 3: Distribution of margin (referred as *improvement*) between the undistilled and BERT-base distilled with `ARMADA` and Stable Diffusion teacher model for different values of $\alpha$, $\beta$ and $\gamma$. **A.** $\alpha$ highlights the importance of output alignment, **B.** $\beta$ highlights the importance of manifold alignment between `TS Aligner` and the student, and **C.** $\gamma$ highlights the importance of auxiliary output on the combined loss objective.

head. `TS Aligner` benefits even more significantly, with an improvement margin of 6.07%, underscoring the importance of auxiliary outputs in `ARMADA` framework.

| Models | Loss type | CoLA | MRPC | RTE | STS-B | CB | COPA | BoolQ | WiC |
|---|---|---|---|---|---|---|---|---|---|
| | | Mcc | Acc | Acc | Pear | Acc | Acc | Acc | Acc |
| `TS Aligner` | - | | 0.7 | 66.4 | 50.2 | 1.3 | 48.9 | 41.0 | 45.4 | 48.9 |
| | cosine | 59.0 | 83.4 | 70.4 | 88.6 | 87.5 | 64.0 | 74.7 | 57.1 |
| BERT-base | euclid | 60.1 | 84.1 | 67.5 | 88.2 | 85.7 | 63.0 | 73.9 | 55.2 |
| | elementwise | 58.5 | 82.3 | 65.7 | 88.6 | 85.7 | 65.0 | 74.4 | 57.2 |

Table 10: Auxiliary output results for `TS Aligner` and BERT-base student model with different $\mathcal{L}_{manifold}$ loss objectives on GLUE-small and SuperGLUE tasks.

**Ablation study.** Our in-depth analysis of the BERT-base model's performance across various GLUE-small and SuperGLUE NLU tasks under different hyperparameter settings uncovers interesting insights into the optimization of cross-modal distillation. As illustrated in Figure 3, our exploration into the effects of the hyperparameters, $\alpha$, $\beta$, and $\gamma$, reveals fine-grained understandings of their impact on cross-modal distillation efficacy. Notably, setting $\alpha$ to 0, thereby excluding the logit matching soft loss from the final training loss, yields a consistent 0.6% improvement across all tasks compared to an $\alpha$ value of 0.5, which gives equal importance to output loss and soft loss. This finding suggests that prioritizing the output loss can lead to better distillation outcomes. The significance of $\beta$, which regulates the weight of the manifold projection

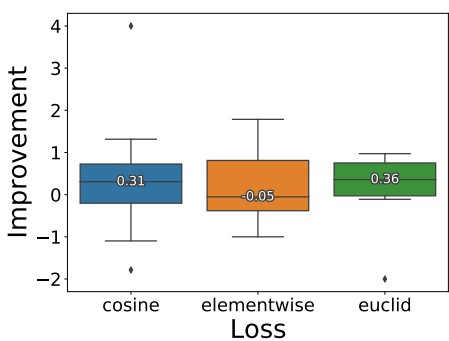

Figure 4: Performance improved in distilled BERT-base model over undistilled variant across all tasks for different $\mathcal{L}_{manifold}$ functions.

loss, is underscored by its performance enhancement in seven out of eight tasks when set to 1, indicating a positive association between higher $\beta$ values and improved model performance. Conversely, $\gamma$, which determines the weight assigned to the auxiliary output loss, demonstrates a variable influence on the effectiveness of cross-modal distillation. A $\gamma$ value of 1 enhances performance by 0.85%, significantly higher than $\gamma = 0$ (improvement 0.36%), highlighting the role of auxiliary losses in the distillation process.

Similar ablation analyses with different manifold projection losses in Figure 4 suggest that all three manifold-alignment loss objectives effectively distil knowledge from the multimodal teacher model to language-only

student models, with $\mathcal{L}_{euclid}$ being most effective and robust on average (median improvement of 0.36 as opposed to 0.31 of $\mathcal{L}_{cosine}$ and $-0.05$ of $\mathcal{L}_{elementwise}$). As explained in Section 3, all the loss objectives enforce different regularizations for the student. We hypothesize that by drawing a balance between the regularization and the task objective, $\mathcal{L}_{euclid}$ draws a balance between the other two losses, leading to the most significant performance gain.

| Model | CoLA | MRPC | RTE | STS-B | CB | COPA | BoolQ | WiC |
|---|---|---|---|---|---|---|---|---|
| | Mcc | Acc | Acc | Pear | Acc | Acc | Acc | Acc |
| `TS Aligner` | 9.9 | 66.5 | 54.5 | 2.1 | 51.8 | 54/0 | 53.9 | 52.3 |
| `ARMADA` with $\mathcal{L}_{cosine}$ | 59.3 | 84.3 | 69.7 | 89.3 | 85.7 | 62.0 | 73.2 | 57.2 |
| `ARMADA` with $\mathcal{L}_{euclide}$ | 59.3 | 82.4 | 69.7 | 89.3 | 85.7 | 54.0 | 76.2 | 56.4 |
| `ARMADA` with $\mathcal{L}_{elementwise}$ | 59.3 | 82.4 | 69.7 | 89.3 | 83.9 | 55 | 76.7 | 55.6 |

Table 11: Results of `TS Aligner` and BERT-12L with `ARMADA` on all GLUE-small and SuperGLUE tasks with higher-capacity `TS Aligner`. We increase the number of hidden layers in `TS Aligner` and remove the projection mapping $\tilde{F}_{ts}$ and auxiliary output head $O^{\mathcal{T}}_{aux_{ts}}$. The objective of this experiment is to corroborate that performance improvement by `ARMADA` is not entirely due to the capacity of the aligner model.

A natural concern is that the gains observed with `ARMADA` may simply arise from increased parameter capacity rather than from structured cross-modal alignment. To directly test this hypothesis, we construct a capacity-matched variant in which we increase the number of hidden layers in `TS Aligner` to match (and slightly exceed) the total parameter count introduced by the projection module $\tilde{F}_{ts}$ and the auxiliary output head $O^{\mathcal{T}}_{aux_{ts}}$, while removing both alignment components. This setup preserves (and in fact increases) model capacity but eliminates the cross-modal projection and auxiliary supervision mechanisms. Table 11 reports results for BERT-12L on GLUE and SuperGLUE under this higher-capacity `TS Aligner` configuration. Contrary to the capacity-expansion hypothesis, we observe an average 2.8% performance drop relative to the full `ARMADA` configuration, with the degradation being statistically significant ($p < 0.05$). Notably, this variant does not recover the $+1$–$3\%$ gains previously observed with proper alignment, despite having comparable or greater parameter count. These findings demonstrate that performance improvements are not attributable to additional depth or parameter expansion. Instead, effective cross-modal knowledge distillation critically depends on the structured projection mapping and auxiliary alignment objectives, which facilitate semantically meaningful teacher–student correspondence. Capacity alone, in the absence of alignment, is insufficient to produce the observed gains.

## 5.2 Results on Instruction-tuning Tasks

| Teacher | Distillation Type | Dolly | SNI | SelfInst | UNI | Vicuna |
|---|---|---|---|---|---|---|
| - | undistilled | 31.30 | 26.68 | 18.44 | 32.01 | 17.59 |
| language-only | SeqKD | 31.45 | 28.48 | 18.61 | 31.05 | 17.74 |
| text-to-image | `ARMADA` with $\mathcal{L}_{cosine}$ | **31.94** | **27.13** | **19.83** | **33.57** | **18.73** |
| | `ARMADA` with $\mathcal{L}_{euclid}$ | 30.57 | **28.25** | 18.06 | 31.59 | **18.15** |
| | `ARMADA` with $\mathcal{L}_{elementwise}$ | **31.38** | 26.68 | 18.28 | **32.27** | **18.16** |
| text-to-video | `ARMADA` with $\mathcal{L}_{cosine}$ | **33.02** | **26.98** | **18.49** | **32.07** | **18.44** |
| | `ARMADA` with $\mathcal{L}_{euclid}$ | **33.07** | **27.75** | **18.96** | 31.51 | **17.92** |
| | `ARMADA` with $\mathcal{L}_{elementwise}$ | **31.85** | 26.57 | **18.84** | **32.01** | **17.84** |
| text-to-audio | `ARMADA` with $\mathcal{L}_{cosine}$ | **32.95** | **27.46** | **18.92** | **33.67** | **18.15** |
| | `ARMADA` with $\mathcal{L}_{euclid}$ | **32.68** | **27.70** | **18.66** | **32.70** | **18.01** |
| | `ARMADA` with $\mathcal{L}_{elementwise}$ | **32.81** | **28.07** | **19.57** | **33.01** | **18.67** |

Table 12: Results on instruction-tuning tasks with LLaMA-3.2-3B with different multimodal teacher models. We also highlight the RougeL score when distilled from language-only LLaMA-8B model with SeqKD (Kim & Rush, 2016). A p-value of 0.005 with one-sided t-test suggests that `ARMADA` provides better post-distillation performance than SeqKD. Interestingly, the teacher models used with `ARMADA` are significantly smaller ($<$2B) than the language-only teacher model (8B) used in SeqKD.

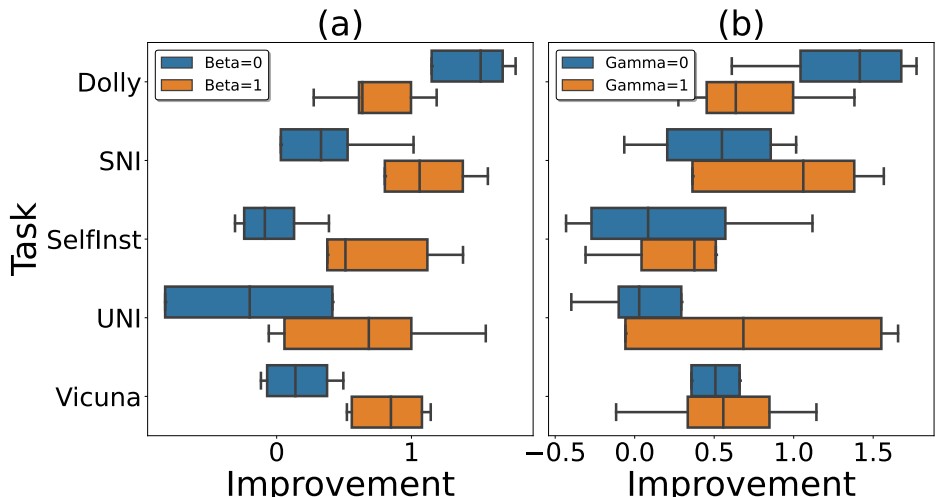

Figure 5: Distribution of improvement between the undistilled and `ARMADA`-distilled LLaMA-3B, for different values of $\beta$ and $\gamma$. A two-way ANOVA using ordinary least square (OLS) regression model confirms the statistical significance (p-value $7e-5$ and $5e-3$ for $\beta$ and $\gamma$, respectively) of both the hyperparameters.

Table 12 highlights the performance of LLaMA-3.2-3B-Instruct distilled from different multimodal teacher models with `ARMADA`. Across all five tasks, `ARMADA` demonstrates consistent performance improvement over the undistilled LLaMA-3B, with maximum gains of 1.8%, 1.6%, and 1.5% on Dolly, UNI, and SNI tasks, respectively. Surprisingly, `ARMADA` also outperforms SeqKD (unimodal distillation from LLaMA-8B teacher) by over 1% margin in several tasks like Dolly, SelfInst, and UNI, albeit using significantly smaller teacher models. Among the different modalities, we observe the highest average improvement of 1.4% with the text-to-audio teacher model, followed by 1.3% improvement by the text-to-image teacher model. However, ANOVA tests suggest that there might not be any apparent difference between the different teacher modalities in terms of the effectiveness of the LLaMA-3B student model.

Figure 5 highlights the importance of the manifold alignment weight $\beta$ and auxiliary output weight $\gamma$. Except for Dolly evaluation tasks, we observe 0.5% additional benefit in other instruction-tuning tasks due to both hyperparameters. Moreover, ANOVA tests suggest that both the hyperparameters are statistically important (p-value $< 0.01$) for the higher performance of `ARMADA`. Figure 6 illustrates the impact of different manifold alignment loss objectives on the student model. On instruction-tuning tasks, under $\mathcal{L}_{\text{cosine}}$, `ARMADA` exhibits the highest performance (average improvement of 0.9%), followed by both $\mathcal{L}_{\text{elementwise}}$ and $\mathcal{L}_{\text{euclid}}$ (0.5%). However, ANOVA test results suggest that all the loss objectives are equally effective (as we fail to reject the null hypothesis stating that the loss functions are all equally performant under a significance level of 5%).

| Teacher | Distillation Type | Dolly | SNI | SelfInst | UNI | Vicuna |
|---|---|---|---|---|---|---|
| - | undistilled | 35.34 | 28.19 | 18.01 | 32.78 | 17.85 |
| text-to-image | ARMADA with $\mathcal{L}_{cosine}$ | 33.67 | **28.77** | **19.36** | 32.59 | **18.89** |
| | ARMADA with $\mathcal{L}_{euclid}$ | 33.71 | 28.11 | **19.14** | **34.53** | 18.72 |
| | ARMADA with $\mathcal{L}_{elementwise}$ | 34.63 | **29.06** | **19.80** | 32.54 | **18.16** |
| text-to-video | ARMADA with $\mathcal{L}_{cosine}$ | 33.53 | **29.28** | 18.57 | **33.80** | **18.82** |
| | ARMADA with $\mathcal{L}_{euclid}$ | 32.02 | 28.07 | **21.31** | 32.41 | **18.94** |
| | ARMADA with $\mathcal{L}_{elementwise}$ | 30.44 | 27.56 | **18.97** | 32.76 | **18.36** |
| text-to-audio | ARMADA with $\mathcal{L}_{cosine}$ | 33.14 | **29.63** | **20.08** | **34.35** | **19.96** |
| | ARMADA with $\mathcal{L}_{euclid}$ | 34.78 | 27.26 | 17.47 | 32.84 | **19.51** |
| | ARMADA with $\mathcal{L}_{elementwise}$ | 34.67 | **29.13** | **18.73** | **34.22** | **18.18** |

Table 13: Results on instruction-tuning tasks with LLaMA-3.1-8B with different multimodal teacher models.

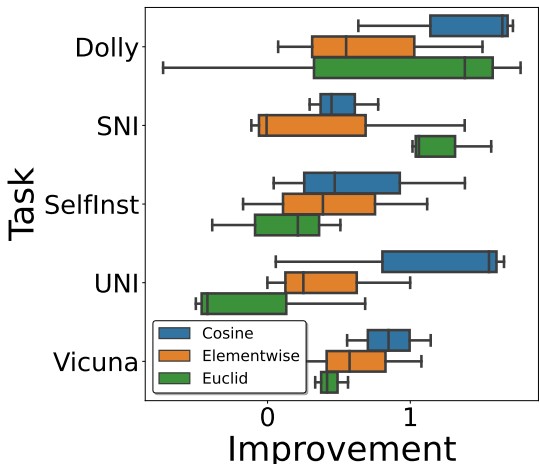

Figure 6: Distribution of improvement between the undistilled and `ARMADA`-distilled LLaMA-3B, for different manifold alignment loss functions. A p-value of 0.52 with the ANOVA test suggests that we may not reject the null hypothesis of dissimilarity between the different loss functions. Therefore, the empirical evidence suggests that all the loss functions could be equally effective in the `ARMADA` framework.

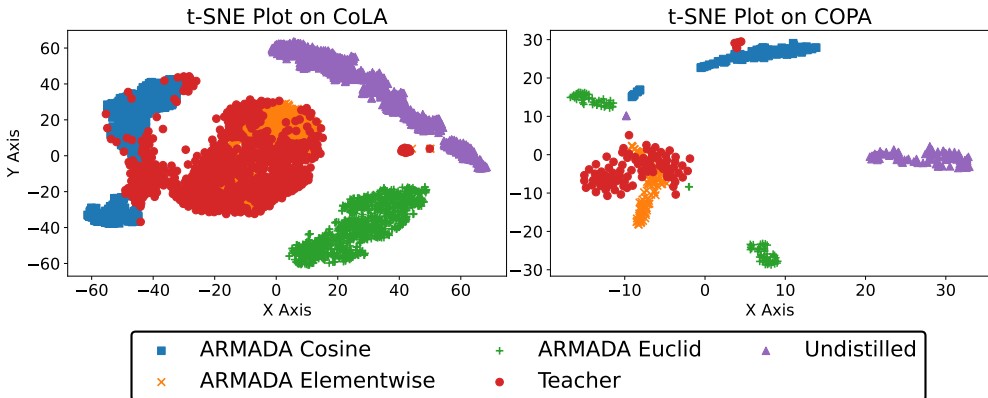

Figure 7: t-SNE plots of embeddings obtained from different models on CoLA and COPA tasks.

Table 13 further highlights the performance of `ARMADA` with a larger student model, LLaMA-8B. Except for Dolly evaluation, in the other four tasks, `ARMADA` turns effective, even for a superior baseline model LLaMA-8B. Strikingly, on tasks like SelfInst and Vicuna, `ARMADA` improves the base model's performance by over 2%. A possible reason behind the muted performance in Dolly evaluation could be the narrow prompt diversity and template simplicity, where the undistilled LLaMA-8B model already shows strong performance, reducing the marginal benefit of distilling cross-modal knowledge. In terms of the different teacher modalities, we observe the text-to-video model to be most effective, with the highest performance improvement of 3.3%, followed by the text-to-audio (2.1%) and text-to-image (1.8%) teacher models. However, ANOVA tests still suggest that the teacher modalities might be equally effective, as we fail to reject the null hypotheses.

## 6 Discussion

To gain a deeper understanding of cross-modal distillation on the GLUE-small and SuperGLUE tasks, we conduct further empirical and statistical analyses to answer some pertinent questions.

**How does the cross-modal distillation work?** Proposition 2 suggests that manifold alignment ensures equivalence in the output and auxiliary output spaces. Therefore, a manifold alignment between a student

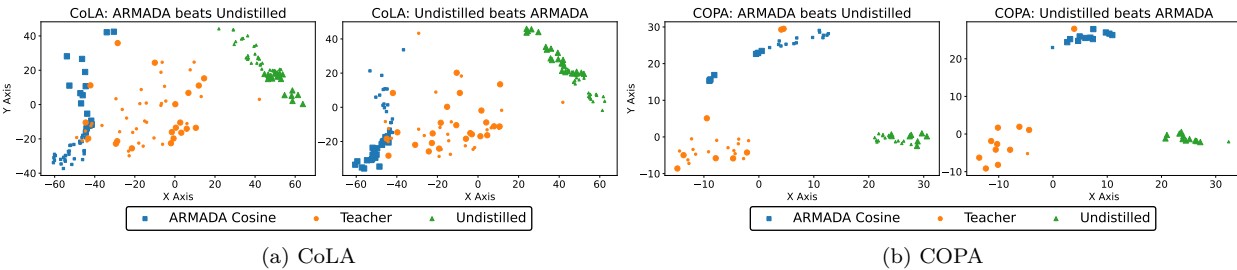

Figure 8: Error analysis on CoLA and COPA tasks. Different output classes are highlighted with different sizes.

model and `TS Aligner` can enforce a regularization for the student model, which allows the student to learn different abstractions and prevents it from overfitting noises in the original modality. Figure 7 highlights the embeddings obtained by different variants of the student and teacher models, projected onto a 2-dimensional space. Due to the most stringent regularization, the student model under element-wise manifold distance projects data points closer to that of the teacher. Contrarily, the undistilled and non-regularized student model projects the data points furthest from the teacher.

To further strengthen our argument, we perform error analysis on two natural language understanding tasks – CoLA and COPA. Figure 8 illustrates the t-SNE embeddings for instances where `ARMADA` beats the undistilled models (we denote as 'scenario 1') and vice versa (denoted as 'scenario 2'). To understand how the cluster structure changes between different models, we report Silhouette scores obtained with `ARMADA` and the undistilled BERT-base model on CoLA and COPA under different scenarios in Table 14.

A higher Silhouette score indicates better semantic alignment between similar instances with similar target outputs. In scenario 1, where `ARMADA` beats the undistilled model, the Silhouette of `ARMADA` improves over the undistilled model by 76%. On the other hand, in scenario 2, the Silhouette score for `ARMADA` drops by 20%, indicating poorer semantic cohesion among semantically similar instances. Table 15 highlights a few selected examples from the

| Methods | CoLA | | COPA | |
|---|---|---|---|---|
| | Scen. 1 | Scen. 2 | Scen. 1 | Scen. 2 |
| Undistilled | 0.31 | 0.40 | 0.31 | **0.59** |
| `ARMADA` | **0.52** | **0.48** | **0.57** | 0.38 |

Table 14: Silhouette scores (higher the better) for different scenarios with undistilled and `ARMADA` models.

CoLA task where we observe that post-distillation, the semantic cohesion improves. The improvement is particularly significant for examples from scenario 1, where better semantic alignment improves performance post-distillation. From these results, we conclude that abstract representations from multimodal teachers allow `ARMADA` to incorporate different signals during representation learning, allowing the model to learn different semantic abstractions (without any explicit mental image) even for linguistically challenging tasks, which improves their generalization capabilities on downstream tasks.

| Scenario | Text | Gold Label | Prediction | Dist. Before | Dist. After (% Change) |
|---|---|---|---|---|---|
| 1 | As you eat the most, you want the least. | 0 | 0 | 0.097 | 0.001 (99%) |
| | Who is she trying to make up to now? | 1 | 1 | 0.575 | 0.007 (99%) |
| 2 | The more does Bill smoke, the more Susan hates him. | 0 | 1 | 0.073 | 0.016 (78%) |
| | Clearly, John probably will immediately learn French perfectly. | 1 | 0 | 0.258 | 0.027 (89%) |

Table 15: Cosine distance from the cluster centroid to the embeddings obtained by BERT-base model before and after distilled with `ARMADA`. Examples selected from CoLA classification task.

To further highlight the regularization effect of cross-modal distillation, we perform a quantitative evaluation on the representations obtained by the BERT-12L student model with and without `ARMADA` for each text input on CoLA and COPA tasks. Towards that, we compute a *cluster purity* score of student model representations to quantitatively assess the impact of `ARMADA` on students' hidden representations. Given a dataset $\mathcal{D} = \{(x_i, y_i)\}_{i=1}^{N}$ where $x_i \in \mathbb{R}^d$ is the latent representation of the $i$-th example and $y_i \in \mathcal{Y}$

| Loss type | CoLA | MRPC | RTE | STS-B | CB | COPA | BoolQ | WiC |
|---|---|---|---|---|---|---|---|---|
| Cosine | **6.75 (0.00)** | **6.95 (0.00)** | **3.33 (0.00)** | **54.63 (0.00)** | **3.89 (0.00)** | 0.49 (0.62) | -0.29 (0.78) | **12.91 (0.00)** |
| Euclid | **6.54 (0.00)** | **8.40 (0.00)** | **3.28 (0.00)** | **51.33 (0.00)** | **3.20 (0.00)** | 1.47 (0.14) | -3.20 (0.00) | **16.29 (0.00)** |
| Elementwise | **6.42 (0.00)** | **8.03 (0.00)** | **3.16 (0.00)** | **51.51 (0.00)** | **4.48 (0.00)** | **5.85 (0.00)** | -2.45 (0.01) | **15.81 (0.00)** |

Table 16: $T$-statistic ($p$-value) between student training loss distributions without and with `TS Aligner` training. A positive $t$-statistics (highlighted in bold) indicates that the expected student loss without a trained `TS Aligner` is higher than the student loss with a jointly trained `TS Aligner` model, indicating the importance of joint aligner-student training.

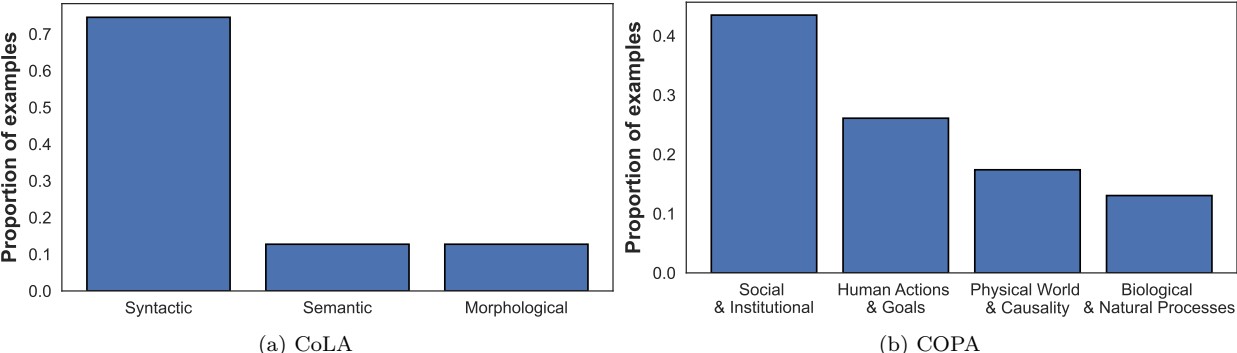

(a) CoLA        (b) COPA

Figure 9: Categorization of examples where BERT-12L distilled with `ARMADA` predicts correctly and the undistiled variant fails.

is its corresponding ground-truth correctness label, let a clustering algorithm assign each $x_i$ to a cluster $c_i \in \mathcal{C} = \{1, 2, \ldots, K\}$. The cluster purity is then defined as:

$$\text{Purity} = \frac{1}{N} \sum_{k=1}^{K} \max_{y \in \mathcal{Y}} |\{x_i \mid c_i = k \wedge y_i = y\}|.$$

This metric evaluates the extent to which a single class label dominates each cluster, and therefore serves as a proxy for the alignment between the internal structure of the latent space and model correctness. A higher purity score indicates better separability between correct and incorrect predictions in the representation space, reflecting stronger semantic abstraction and regularization.

For the undistilled BERT-12L, we obtained purity scores of 0.77 and 0.56 on CoLA and COPA tasks, respectively (We use KMeans clustering with $K = 2$). With `ARMADA`, however, we obtain purity scores of 0.81 and 0.68, respectively. These results and the semantic cohesion results highlighted in the previous section suggest that the cross-modal teacher induces more topologically organized and semantically meaningful representations in the student model. This highlights that cross-modal distillation improves task performance and guides the student model to form more structured, discriminative, and generalizable internal representations.

To provide a concrete characterization of where cross-modal distillation yields improvements at the individual example level, we manually analyze test instances where the BERT-12L student distilled with `ARMADA` predicts correctly while the undistilled baseline fails. We focus on CoLA and COPA, as they probe distinct linguistic and conceptual phenomena.

For CoLA (linguistic acceptability), we categorize such examples into three broad classes: Syntactic, Semantic, and Morphological. Syntactic examples involve structural dependencies and long-distance constraints (e.g., "Who did John send the book?", testing filler–gap dependencies). Semantic examples violate plausibility or selectional restrictions (e.g., "The bookcase ran", where the subject is incompatible with the verb). Morphological examples involve agreement or inflectional mismatches (e.g., "The cat were bitten by the dog", singular–plural disagreement).

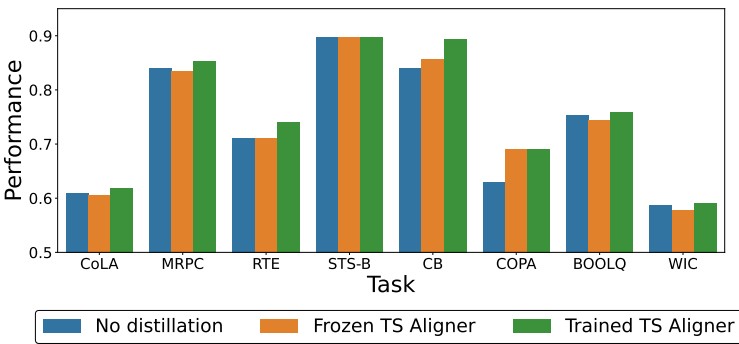

Figure 10: Performance of distilled BERT-base under different training curricula. The performance improvement with a trained `TS Aligner` demonstrates the importance of `TS Aligner` training on cross-modal distillation.

As shown in Figure 9(a), improvements are highly non-uniform: over 70% of corrected cases fall into the syntactic category, whereas semantic and morphological phenomena each account for roughly 12–13%. This indicates that gains on CoLA are concentrated on structurally complex constructions rather than uniformly distributed across error types.

For COPA (causal and commonsense reasoning), we categorize improved examples into four classes: Social & Institutional, Human Actions & Goals, Physical World & Causality, and Biological & Natural Processes. Social & Institutional cases involve culturally grounded or socially structured knowledge (e.g., "The man dressed in his best suit" → cause: "He scheduled a meeting with an important client"). Human Actions & Goals capture intentional behavior. Physical World & Causality involve object interactions and physical constraints. Biological & Natural Processes reflect physiological or natural events (e.g., "I coughed" → cause: "I inhaled smoke").

Figure 9(b) shows that approximately 42% of the corrected COPA examples belong to the Social & Institutional category, followed by Human Actions (26%), Physical World (17%), and Biological processes (15%). This skew suggests that `ARMADA` disproportionately benefits examples requiring socially grounded commonsense knowledge, rather than uniformly improving all causal categories.

Taken together, this analysis demonstrates that cross-modal distillation does not yield homogeneous gains. Instead, improvements concentrate in linguistically complex (syntactic) constructions for CoLA and socially grounded commonsense reasoning for COPA. This non-uniform distribution provides concrete evidence that `ARMADA` transfers structured semantic and conceptual priors, rather than acting as a generic regularizer.

**How does the multimodal teacher impact the student?** Table 6 highlight the challenges faced by `TS Aligner`, which, being relatively shallow (only $0.8M$ learnable parameters), struggles with most NLU tasks under different teacher models. Despite this, it remarkably distils knowledge to student models, even from a position of lower performance, as evidenced in tasks like CoLA and STS-B. To understand the impact of `TS Aligner` on the distilled student model, we analyze the students' performances under a frozen alignment module. Under this setup, `TS Aligner` is randomly initialized and not trained further.

Figure 10 highlights that in six of eight NLU tasks the student model distilled with the frozen `TS Aligner` is inferior to the student model without distillation. However, the performance improves significantly with a trained `TS Aligner` model. This observation asserts that the aligner must first learn a task to ensure better and more effective distillation. Even if the aligner is poor and cannot perform well on a task, the module's learning curve enriches the student's performance. Table 16 highlights the $t$-statistic between the student output loss $\mathcal{L}_{\mathcal{T}}(Y^{\mathcal{T}}, O_s^{\mathcal{T}})$ empirical distributions without and with aligner training. A strong positive $t$-statistics with a low $p$-value indicates the positive impact of a trained `TS Aligner` module on students' learning. The empirical evidences described in Section 5 suggests that vision-language models implicitly encode abstract knowledge that can benefit pure language models, despite the absence of explicit textual representations. This raises intriguing questions about the nature of multimodal knowledge representation and how different modalities contribute to generalizability in language models.

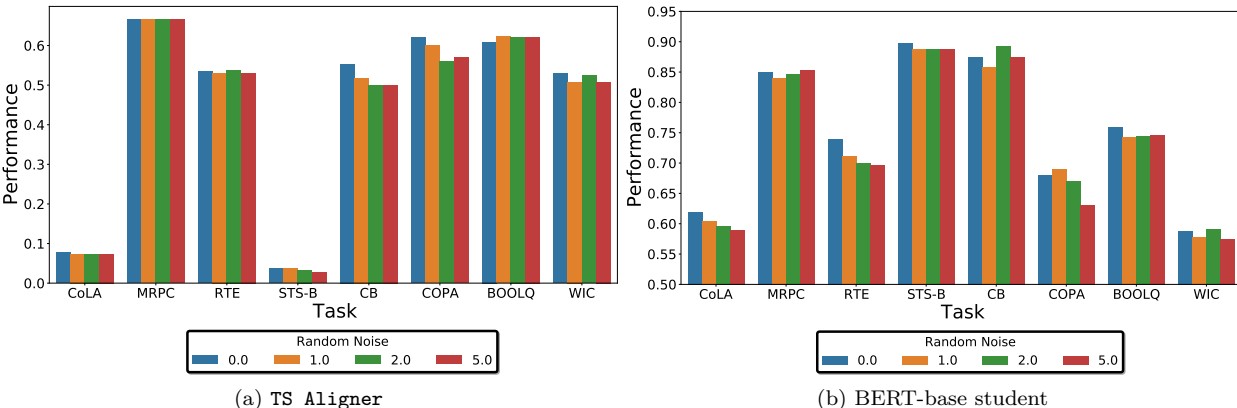

Figure 11: Performance of `TS Aligner` and distilled BERT-base student under teacher input noises.

**Do teacher's inputs matter?** The role of teacher's inputs in the cross-modal distillation process is pivotal for our study, examining how variations in the quality of these inputs affect the performance of `TS Aligner` and student models. To probe this, we introduce Gaussian noise $\epsilon \sim \mathcal{N}(0, \sigma)$ to the inputs fed to `TS Aligner` and observe the resultant impact on both the teacher's performance and that of the distilled student model. This approach allows us to simulate real-world scenarios where input data may not always be pristine and to understand the robustness of both models to such perturbations. Figure 11 highlights the performance of `TS Aligner` and the student model for different values of $\sigma = \{0, 1, 2, 5\}$. We generally observe that the aligner is more robust to the input noise than the student model. However, the performances drop only insignificantly with larger values of $\sigma$. To quantitatively assess the impact of input noise on model performance, we calculate a sensitivity score defined as $\frac{Var(\text{performance})}{Var(\sigma)}$. This metric, illustrated in Figure 12, directly compares how performance variability relates to input noise variability for both `TS Aligner` and student models. Notably, on tasks such as CoLA, MRPC, and RTE, student models exhibit higher sensitivity scores than the aligner, highlighting a greater vulnerability to the quality of teacher inputs during the distillation process.

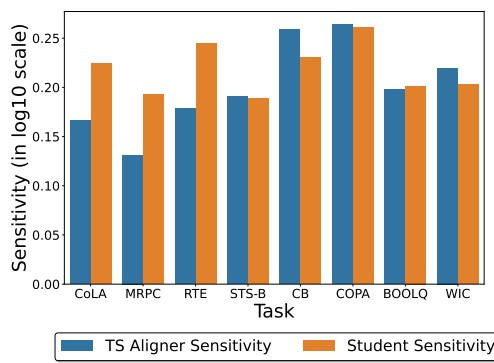

Figure 12: Sensitivity of `TS Aligner` and distilled BERT-base student models under input noises.

Even for a larger student model like LLaMA-3B, the impact of teacher representation quality on downstream instruction-tuning tasks is surprisingly strong (see Figure 13). At higher noise levels, corresponding to poorer teacher representation quality, the performance of the distilled student degrades significantly, often approaching that of the undistilled model. These behaviors complement the findings of Ramesh et al. (2025), who observed diminished student performance when distillation was conducted using low-quality teacher inputs. Despite this drop, the performance of the student remains comparable to the undistilled baseline, highlighting the regularization effect of our cross-modal distillation mechanism: when the teacher signal quality degrades, its influence on the student diminishes naturally. This reveals a critical distinction from unimodal knowledge distillation approaches, which often struggle with balancing fidelity and generalization be-

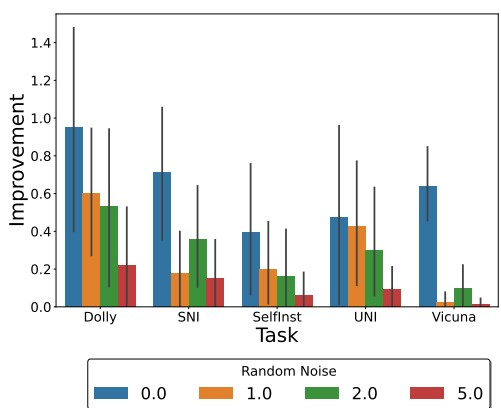

Figure 13: Improvement of `ARMADA` over undistilled LLaMA-3B for different teacher input noises. Higher noise indicates poorer quality of teacher representations.

tween teacher and student (Ramesh et al., 2025; Cha & Cho, 2025). In contrast, cross-modal knowledge distillation leverages abstract semantic representations and implicit regularization, enabling robust transfer of knowledge without requiring explicit signal fidelity.

| Model | CoLA | MRPC | RTE | STS-B | CB | COPA | BoolQ | WiC |
|---|---|---|---|---|---|---|---|---|
| | Mcc | Acc | Acc | Pear | Acc | Acc | Acc | Acc |
| TS Aligner | 4.9 | 66.5 | 54.9 | 2.5 | 41.1 | 54.0 | 55.0 | 53.0 |
| ARMADA with $\mathcal{L}_{cosine}$ | 57.9 | 84.4 | 71.5 | 90.0 | 81.1 | 63.0 | 75.8 | 56.4 |
| ARMADA with $\mathcal{L}_{euclide}$ | 59.4 | 84.5 | 70.8 | 89.5 | 82.1 | 63.0 | 75.5 | 57.0 |
| ARMADA with $\mathcal{L}_{elementwise}$ | 60.4 | 84.8 | 70.0 | 89.8 | 83.9 | 63.0 | 76.7 | 57.4 |

Table 17: Results of `TS Aligner` and BERT-12L with `ARMADA` on all GLUE-small and SuperGLUE tasks with shuffled teacher inputs. We randomly shuffle the image embeddings obtained by the Stable Diffusion teacher model during training the student models.

**Does semantic alignment in teacher representations matter?** While Gaussian perturbations probe robustness to representation quality, they do not disrupt the semantic correspondence between teacher inputs and student samples. To explicitly test whether cross-modal distillation relies on meaningful semantic alignment rather than generic regularization, we perform a stronger intervention: we shuffle teacher inputs across samples during training, thereby introducing intentional semantic misalignment while keeping the architecture and losses unchanged.

Table 17 reveals a markedly different behavior from the Gaussian noise experiments. Under shuffled inputs, `TS Aligner` performance drops by 3.8% (p = 0.08, not statistically significant), whereas the BERT-12L student degrades by 1.8% (p = 0.03, statistically significant). More critically, on semantically complex tasks such as BoolQ, performance decreases by up to 5%, substantially larger than the marginal variations observed under Gaussian noise, and, more critically, drops below the undistilled model's performance. Unlike noisy-but-aligned inputs, shuffled representations destroy the structured correspondence between text and teacher signal, eliminating the gains observed in standard `ARMADA` training.

This contrast establishes an important distinction: robustness to Gaussian perturbations reflects tolerance to moderate representation degradation, whereas shuffled inputs invalidate the semantic signal entirely and significantly harm the student. Consequently, the improvements under normal training cannot be attributed to generic regularization or architectural smoothing effects. Instead, they require structured, semantically aligned cross-modal information from the teacher. Together with the noise sensitivity analysis above, these results demonstrate that cross-modal distillation operates through meaningful semantic transfer, while naturally attenuating its influence when teacher representations become unreliable.

## 7  Conclusion

This paper described `ARMADA`, a framework for knowledge distillation from vision-language teacher models to language-only student models. Our analytical results highlighted the importance of cross-modal knowledge transfer, even for large pre-trained language models. Although the empirical evaluations primarily focused on the NLU tasks, the underlying principles extend far beyond these areas. The framework's adaptability and generality suggest that it could be seamlessly applied to a broader spectrum of tasks, potentially revolutionizing how models learn from disparate data sources. The findings challenge traditional assumptions about modality-specific learning and paves the way for efficient, scalable, and architecture-agnostic knowledge transfer across modalities. Given the increasing prominence of large pre-trained models in achieving state-of-the-art results across numerous domains, understanding how cross-modal distillation further refines their abilities in language understanding and reasoning, presents a fertile ground for future investigations.

## Limitations

This work challenges traditional assumptions in knowledge distillation by demonstrating that vision-language models can significantly enhance language-only models, paving the way for efficient, multimodal AI systems

without additional computational overhead. The findings have broad implications for resource-efficient AI, enabling scalable improvements in NLP without requiring direct access to large-scale text-only corpora. However, care must be taken to ensure that cross-modal knowledge transfer does not introduce unintended biases from vision models into language models, emphasizing the need for fairness and interpretability in future research.

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

# A  Datasets

## A.1  Natural Language Understanding

The General Language Understanding Evaluation (GLUE)[‡] benchmark (Wang et al., 2018) and SuperGLUE[§] benchmark (Wang et al., 2019) evaluate the language understanding capabilities of language models. We elaborate on the GLUE and SuperGLUE tasks as follows:

**CoLA** (Warstadt et al., 2019), or The Corpus of Linguistic Acceptability, comprises acceptability judgments from books and journal articles. The task is to indicate the grammatical correctness of the given sentence as "acceptable" or "unacceptable" using the Matthews correlation coefficient as the evaluation metric.

**MRPC** (Dolan & Brockett, 2005), Microsoft Research Paraphrase Corpus comprises pairs of sentences extracted from online news sources. The objective is to predict whether the provided pair of sentences are paraphrases of each other or not.

**RTE** (Dagan et al., 2005; Haim et al., 2006; Giampiccolo et al., 2007; Bentivogli et al., 2009), short for Recognizing Textual Entailment, are formed by combining a series of annual textual entailment challenges. The task comprises categorizing whether the two sentences entail each other.

**STS-B** (Cer et al., 2017), or Semantic Textual Similarity, compiled from various sources, including news headlines, video-image descriptions, and NLI descriptions. The task is to predict the degree of similarity between two sentences.

**SST-2** (Socher et al., 2013), Stanford Sentiment Treebank involves a two-class classification task, predicting the sentiment of the given movie review as 'positive' or 'negative'.

**QQP** (Chen et al., 2018), or Quora Question Pairs, is a dataset comprising questions collected from the question-answering platform - Quora. The task involves predicting whether or not the given two questions are duplicates of each other.

**MNLI** (Williams et al., 2017) Multi-Genre Natural Language Inference involves a three-class classification problem. The goal is to predict whether the premise "entails"/"contradicts" or is neutral towards the hypothesis statement.

**QNLI** (Wang et al., 2018) or Question Answering NLI is a dataset derived from the Stanford Question Answering Dataset. The task comprises a question and a sentence. The goal is to predict whether or not the given sentence contains the answer to the question.

**BoolQ** (Clark et al., 2019), or Boolean Questions, involves binary inquiries sourced from the Google search engine. These questions are combined with pertinent paragraphs extracted from Wikipedia articles, ensuring that the provided paragraphs contain accurate answers to the queries.

---

[‡]`https://gluebenchmark.com/`
[§]`https://super.gluebenchmark.com/`

**CB** (De Marneffe et al., 2019), short for CommitmentBank, comprises concise texts featuring embedded clauses. These examples are extracted from diverse sources, including the British National Corpus Fiction and Wall Street Journal. The task comprises of three-class textual entailment, where each example consists of a premise and a corresponding hypothesis and is labeled as either "contradiction," "neutral," or "entailment".

**COPA** (Roemmele et al., 2011), abbreviated for Choice of Plausible Alternatives, is a causal reasoning task with the goal of selecting the most plausible choice between two choices given a premise and cause/effect. The examples are sourced from blogs and a photography-related encyclopedia.

**WiC** (Pilehvar & Camacho-Collados, 2018), short for Word-in-Context (WiC), is a word sense disambiguation task that involves binary classification of sentence pairs. Within this task, two text snippets are presented, each featuring a word with multiple potential meanings. The objective is to determine whether the specified word holds the same meaning in both sentences.

## A.2  Multimodal Tasks

Multimodal IMDb (Arevalo et al., 2017) is a collection featuring movies complete with their plot summaries, posters, genres, and an additional set of 50 metadata fields. The dataset is constructed using IMDb IDs from the Movielens dataset.

## A.3  Arithmetic Reasoning

**MultiArith** (Roy & Roth, 2016) consists of arithmetic word problems presented in natural language. The dataset includes various problems that require interpreting and performing basic arithmetic operations.

**SingleEq** (Koncel-Kedziorski et al., 2016) or Single-equation, consists of word problems with single equations of varying length.

**AQuA** (Ling et al., 2017) short for Algebra Question Answering with Rationales, consists of algebraic word problems with natural language rationales.

## A.4  Commonsense Reasoning

**Hellaswag** (Zellers et al., 2019) is a Natural Language Inference (NLI) task which involves finishing the sentence given context. The dataset consists of multiple-choice questions where the task is to choose the most plausible continuation of a given sentence.

**Winogrande** (Sakaguchi et al., 2021) is a commonsense reasoning task inspired by the WSC (Levesque et al., 2012). The goal of the task is to choose the correct choice given two choices.

**ARC-c** & **ARC-e** (Clark et al., 2018) comprise the AI2 Reasoning Challenge (ARC) partitioned into easy (ARC-e) and challenging set (ARC-c). The dataset consists of grade-school science questions in multiple choice format.

**OpenBook-QA** (Mihaylov et al., 2018) consists of elementary level science questions in form of multiple choices requiring additional commonsense knowledge.

## A.5  Instruction-tuning

**Dolly** Gu et al. (2024): We use a curated subset of the `databricks-dolly-15K` dataset, comprising approximately 12.5k training samples, 1k validation samples, and 500 test samples.

**SelfInst** Wang et al. (2022a) dataset includes 252 instruction-following examples designed around user-centric tasks.

**Vicuna** Chiang et al. (2023) contains 80 high-difficulty queries generated by GPT-4, which are employed for evaluating the Vicuna model.

**SNI** Wang et al. (2022b) derived from the Super-Natural Instructions benchmark, this dataset includes about 9,000 examples spanning 119 tasks. Following the methodology in Gu et al. (2024), we segment the data based on the length of reference answers and use the subset with response lengths greater than 10 tokens, which contains roughly 1,600 samples.

**UNI** Honovich et al. (2023) is shortlisted from the Unnatural Instructions dataset, we evaluate on the subset with reference responses longer than 10 tokens, using the first 2,000 samples from this segment.

## B   Baselines

This section elaborates on the unimodal and multimodal knowledge distillation frameworks we use in this paper for evaluating against `ARMADA`.

### B.1   Unimodal KD Baselines

**KD:** Hinton et al. (2015) introduced the concept of distilling knowledge from a bigger complex model to a shallower model by utilizing the *dark knowledge* or the soft label distribution of the teacher model.

**SeqKD:** Motivated by Hinton et al. (2015), Kim & Rush (2016) proposed sequential KD, that forces the student model to generate entire sequences that match the teacher's outputs. This distillation method is particularly useful for sequence generation tasks.

**DML:** Zhang et al. (2018) proposed a mutual learning setup where student models guide each other's learning. The method involved matching the probability distribution of other students by minimizing their KL divergence and task-specific cross-entropy loss.

**PKD:** Sun et al. (2019) proposed distilling knowledge from the intermediate layers of BERT (Devlin et al., 2018) and introduced two different schemes, *PKD-last* and *PKD-skip* for distillation, where the teacher representations are transferred to the student from its last and alternate $k$ layers, respectively.

**TinyBERT:** Jiao et al. (2019) proposed general and task-specific distillation to a 4 layer BERT from BERT-base (Devlin et al., 2018) through attention and hidden states distillation of transformer-based teacher to student.

**TAKD:** Mirzadeh et al. (2020) argued that sub-optimal distillation occurs when a large gap exists between teacher and student. To address this issue, intermediate teachers first distil their knowledge from the teacher model and, in turn, distil their knowledge to the student model.

**MetaDistil:** Zhou et al. (2021) proposed meta-knowledge distillation, different from the previous settings involving a fixed teacher distilling knowledge to the student. This method proposed providing a continuously updated meta-teacher through a bi-level optimization (Finn et al., 2017) for better knowledge transfer.

### B.2   Multimodal KD Baselines

**Vokenization:** Tan & Bansal (2020) proposed visually supervising a language model through "vokens", which are visual representations of tokens. The "vokenization" process involves assigning a relevant image to each token. In addition to the traditional MLM (Devlin et al., 2018), the language model is pre-trained with an additional voken-classification (voken-cls) objective.

**VidLanKD:** Tang et al. (2021) proposed improving language understanding through video-distilled knowledge transfer. The teacher model is pre-trained on a video-text dataset using contrastive learning. Knowledge is then transferred to the student using a variety of KD objectives, including NST (Huang & Wang, 2017) and CRD (Tian et al., 2019).

**Visual Guidance:** Zhang et al. (2021) proposed enhancing word representations by incorporating visual information. The method used a word-image dictionary and an attention mechanism to create text-conditioned image representations and a gating mechanism to combine these with the text representations, improving the original embeddings.

## C  Experimental Settings

For all the tasks, we employ an online distillation (Anil et al., 2018; Zhang et al., 2020) framework, where we train both teacher and student simultaneously. We use $\alpha \in \{0, 0.5\}$, $\beta \in \{0, 1\}$ and $\gamma \in \{0, 1\}$. We use $\tau = 5$ in Equation 2. We use Adam optimizer with weight decay (Loshchilov & Hutter, 2017) for training all the models. For GLUE and SuperGLUE tasks, we use a maximum sequence length of 128 and train both the teacher and the student models for 10 epochs. For GLUE tasks, we use a batch size of 32 and a teacher learning rate of $3e - 3$. For SuperGLUE, we use the batch size as 8 and `TS Aligner` learning rate as $1e - 3$. The BERT student models are fine-tuned for all the GLUE and SuperGLUE tasks with a learning rate of $2e - 5$. DeBERTa and OPT student models are trained with learning rate $3e - 6$. For MM-IMDb classification task, we use Resnet50 (He et al., 2016) pretrained on the ImageNet (Deng et al., 2009) dataset to extract the `TS Aligner` inputs. For the multimodal task, we use the BERT-base model as the student model with only the movie plot summaries as the input and trained it for 20 epochs with a learning rate of $2e - 5$. For generative evaluations, we fine-tune the LLaMA-7B model (Touvron et al., 2023) with LoRA adapter (Hu et al., 2021) rank=8, respectively on Commonsense8K and Math10K datasets curated by Hu et al. (2023) and evaluated on the downstream tasks zero-shot. The model is trained for 3 epochs with a learning rate of $3e - 4$ and batch size of 4.

One Tesla V100 and A100 GPUs were used for conducting the experiments. Each training step for the student and `TS Aligner` model takes 0.16 and 0.02 seconds, respectively, for a batch size of 32.

