# OpenReview forum: "From Images to Words: Efficient Cross-Modal Knowledge Distillation to Language Models from Black-box Teachers"
_TMLR — Rejected by TMLR_

### Review · Reviewer_bGb1 · 2025-12-20

**Summary Of Contributions:**

This paper proposes ARMADA, a cross-modal knowledge distillation framework that leverages frozen multimodal generative models (e.g., text-to-image, text-to-video, and text-to-audio models) as teachers to improve language model training without requiring explicit multimodal supervision. The core idea is to align student language model representations with abstract latent manifolds derived from multimodal teachers via a learned task-specific aligner, supplemented by auxiliary output heads and manifold-level regularization.

## Strengths

1. This paper investigates a novel problem: distilling knowledge from multimodal models into unimodal models can also improve the performance of unimodal models, and it provides a feasible solution to this problem.

2. It proposes an intuitive and easy-to-understand framework that improves distillation performance through aligning model representations, manifold projection embeddings, and auxiliary output heads.

3. The paper conducts detailed and extensive experiments across different teacher–student architectures, datasets, tasks, and hyperparameter ablations, demonstrating the effectiveness of the proposed embedding alignment method.

---

## Weaknesses

Despite providing a large number of experiments, the paper still has several issues.

### 1. Overly strong and unrealistic theoretical assumptions

For example, in **Proposition 2**, the authors attempt to use a topological / homeomorphism-based chain of arguments to show that if the teacher and student are “equivalent” on the manifold (representation space), then their output spaces (as well as auxiliary output spaces) are also “equivalent.”

However, this argument relies on the following implicit assumptions:

1. the projection matrices are invertible;
2. the output mappings are bijective;
3. there exists a homeomorphism between the teacher and student manifolds.

In practice, these assumptions are difficult to justify.

1. Orthogonal projections map representations onto lower-dimensional manifolds, and their invertibility cannot be guaranteed.
2. Neural network mappings are typically many-to-one (i.e., different inputs may produce identical hidden representations or logits), so the assumption that output mappings are bijective usually does not hold.
3. The paper assumes the existence of a homeomorphism \( f \) between the teacher and student manifolds. However, in real models, the teacher representations are latent spaces of diffusion, video, or audio models, while the student representations are hidden states of language models. Their representation geometry, semantic organization, dimensionality, and noise distribution may differ substantially. It is therefore unclear why such manifolds should be homeomorphic.

---

### 2. Unclear motivation and causal relationship for cross-modal distillation

First, it is unclear why cross-modal knowledge distillation is necessary. A natural question is why the authors choose to distill from text-to-vision models such as Stable Diffusion or Midjourney into language models, rather than from large multimodal models such as GPT-4o or the Qwen-VL/Omni series. It is still unclear what direct relationship text-to-image models have with NLU tasks.

Second, the mechanism of the TS Aligner remains insufficiently clear. The TS Aligner is described as “bridging the teacher–student manifold,” but what exactly does the TS Aligner learn? Does it learn task shortcuts, teacher bias, or dataset statistics? How do we know that the observed gains are not simply due to additional supervision or representational smoothing induced by training the aligner on the same task data?

Although the experimental section demonstrates a correlation between representation structure and performance, it does not establish causality.

---

### 3. Insufficient experimental rigor

#### 3.1 Unfair complexity comparison

For example, in **Tables 2–4**, compared with the undistilled baseline, ARMADA introduces:

- a TS Aligner (an additional network),
- an auxiliary output head,
- and a manifold alignment loss.

However, the paper does not report the additional parameter count or the additional training time, and the baseline models are not equipped with modules of equivalent complexity. It therefore remains unclear whether the performance gains of ARMADA come from cross-modal teachers or simply from additional learnable modules and regularization.

---

#### 3.2 Limited multimodal evaluation

The scale of the multimodal task evaluation (MM-IMDb) is limited. In **Table 9 (Results on MM-IMDb)**:

- only one multimodal classification task is used,
- only BERT-base is evaluated,
- the improvements are modest:
  - Macro-F1: +1.2%
  - Micro-F1: +1.3%.

It is questionable whether a single task is sufficient to support broad claims about the effectiveness of multimodal distillation.

---

#### 3.3 Weak representation analysis

The evidence provided by the t-SNE and Silhouette analyses is weak. **Figures 7 and 8** are difficult to interpret. In general, t-SNE is used to visualize embeddings and analyze their separability to assess whether a model learns better decision boundaries.

If we assume that the embeddings shown here are derived from the data, the performance under the three alignment losses is largely similar. Moreover, there is no clear or significant difference between the proposed method and the undistilled baseline. It is therefore unclear whether such comparisons are meaningful.

---

#### 3.4 Baseline comparison clarity

**Table 7** compares the proposed method with existing knowledge distillation methods. However, the authors do not clearly explain how their method differs from these baselines. In addition, the results in Table 7 suggest that the proposed method does not outperform prior methods on individual tasks, raising questions about the advantages of the proposed approach.

In **Table 8**, when comparing with multimodal distillation methods, it is also unclear whether the experimental settings are comparable, and the proposed method does not appear to have a clear performance advantage.

---

### 4. Writing and presentation issues

The experimental section is overly long, scattered, and redundant. Some reduction and consolidation would be beneficial, retaining only the main research questions and conclusions to improve readability.

**Audience:**

Yes

**Audience Explanation:**

Yes, it describes a method for distilling knowledge from large multimodal models into smaller language models to improve efficiency.

**Claims And Evidence:**

No

**Claims Explanation:**

Although the paper reports extensive experiments, several claims are not convincingly supported. The effectiveness of cross-modal distillation, its causal relationship behind the performance gains, and its advantages over existing knowledge distillation methods need further justification.

**Requested Changes:**

Please see the weakness.

---

> ### Author Response · Authors · 2026-02-15
> **Response to Reviewer bGb1 Comments - Part I**
>
> ## Theoretical assumptions in ARMADA
>
> We thank the reviewer for the careful and technically informed critique. We agree that, if interpreted literally, assumptions such as global invertibility, bijectivity, or homeomorphism between teacher and student manifolds would be unrealistic for modern neural networks. However, this is not the setting in which Proposition 2 operates, and we clarify this explicitly in the response below.
>
> First, the theoretical argument does **not** posit a direct topological equivalence between the full teacher and student representation spaces. All operations in the proposition occur between the **TS aligner-induced subspace** and the **student representation space**, not between the raw teacher manifold and the student manifold. The teacher representation is first mapped into a learned intermediate subspace via $\tilde{F}_{ts}$, and alignment is enforced only within this projected space. Thus, the relevant structural comparison is between the *aligner-transformed teacher subspace* and the student subspace, rather than between heterogeneous high-dimensional latent spaces of diffusion models and language models.
>
> Second, regarding invertibility: the projection mapping $\tilde{F}_{ts}$ is explicitly low-rank and not guaranteed to be invertible. The theoretical discussion uses invertibility only as an idealized device for reasoning about structural preservation. In practice, exact invertibility is unnecessary; it suffices that the mapping preserves task-relevant structure locally. Empirically, removing the projection and alignment components results in statistically significant degradation (−2.8% on average, p < 0.05), demonstrating that structured alignment, not invertibility, drives the gains.
>
> Third, strict bijectivity of neural output mappings is not assumed in practice. The argument is better interpreted in terms of equivalence classes induced by the task loss: representations are considered equivalent if they induce identical predictions. Alignment, therefore, preserves decision-relevant structure rather than establishing one-to-one correspondence between individual hidden states. This relaxed interpretation is consistent with the observed empirical behavior.
>
> Finally, we do not claim a global homeomorphism between full teacher and student manifolds. The intended assumption is local and task-conditioned: there exists a low-dimensional, semantically meaningful subspace in which a continuous, structure-preserving correspondence can be learned via the TS aligner. The shuffled-teacher experiment reinforces this interpretation, when semantic correspondence is destroyed (while architecture and losses remain unchanged), performance drops significantly (−1.8% on average; up to −5% on BoolQ), confirming that meaningful local structure, rather than arbitrary geometric similarity, is necessary.
>
> In summary, Proposition 2 should be read as an explanatory structural argument about alignment between the **aligner-induced teacher subspace and the student subspace**, under local and approximate assumptions. It does not assert global topological equivalence between heterogeneous multimodal latent spaces.
>
> ## Confusion around complexity comparison
>
> We thank the reviewer for raising the complexity comparison concern and clarifying an important structural point of the paper. Tables 2–4 are **not baseline comparison tables**; they report the performance of ARMADA relative to the undistilled student under controlled ablations and task settings. The actual baseline comparisons with existing distillation methods are presented separately in Tables 7, 8, and 12. Therefore, Tables 2–4 should not be interpreted as head-to-head architectural fairness comparisons, but as controlled evaluations of the effect of cross-modal distillation on a fixed student backbone.
>
> Regarding complexity, ARMADA introduces <1% additional trainable parameters (TS aligner + auxiliary head), all of which are discarded at inference. Thus, the deployed model matches the undistilled baseline in both parameter count and latency.
>
> To directly address whether gains arise from added modules or regularization, we conducted two controlled experiments. First, a capacity-matched variant increases aligner depth to match (and exceed) the parameter count of the projection and auxiliary modules but removes the projection mapping and alignment objectives. This variant underperforms ARMADA by −2.8% on average (p < 0.05) and is statistically indistinguishable from the undistilled baseline (p > 0.1). Second, a shuffled-teacher control destroys semantic correspondence while keeping the architecture and losses unchanged; performance drops by 1.8% on average (p = 0.03), with up to 5% degradation on BoolQ.

---

> > ### Author Response · Authors · 2026-02-15
> > **Response to Reviewer bGb1 Comments - Part II**
> >
> > ## Motivation of cross-modal KD
> >
> > First, regarding the necessity of cross-modal distillation and the choice of teacher models: our objective is not to compete with large multimodal LLM distillation (e.g., GPT-4o, Qwen-VL), but to study a strictly harder and underexplored regime - transferring knowledge from *non-linguistic, generative, black-box vision-language teachers* into language-only students. Text-to-image models such as Stable Diffusion or Midjourney are trained on massive image–text corpora and encode rich semantic priors (object–attribute relations, spatial reasoning, commonsense co-occurrence patterns) despite not being optimized for NLU. The central question we investigate is whether such cross-modal generative representations contain reusable semantic structure that can benefit purely textual tasks. Empirically, we observe statistically significant gains across GLUE-small and GLUE-large (p = 0.00), up to +5% on COPA, and +5% on Hateful Memes, despite no shared modality at inference time. Importantly, ARMADA also outperforms SeqKD (a unimodal distillation baseline using a larger language teacher) on instruction benchmarks (Table 12), demonstrating that cross-modal teachers can provide complementary information beyond unimodal supervision.
> >
> > Second, regarding the TS aligner mechanism, the aligner does not solve the downstream task independently, nor does it serve as an auxiliary classifier. It learns a projection that maps teacher representations into a student-compatible subspace where task-relevant structure can be aligned. Crucially, the aligner has <1% additional parameters and is discarded at inference, eliminating capacity-based explanations. We explicitly tested alternative hypotheses: (i) a capacity-matched variant with increased depth but no alignment underperforms ARMADA by −2.8% (p < 0.05); (ii) shuffled-teacher training, where semantic correspondence is destroyed, eliminates gains (−1.8% average; up to −5% on BoolQ; p = 0.03). These results demonstrate that improvements require structured teacher–student correspondence, not merely additional supervision or representational smoothing.
> >
> > Finally, regarding causality: while full causal identification in representation learning is inherently challenging, we perform controlled interventions that go beyond correlation. Removing alignment components consistently degrades performance; destroying semantic alignment via teacher shuffling eliminates gains; and increasing model capacity without alignment does not reproduce improvements. These interventional results provide mechanistic evidence that structured cross-modal alignment contributes causally to performance gains.
> >
> > ## Multimodal evaluation
> >
> > We appreciate the reviewer's concern regarding the scale of multimodal evaluation. We clarify that MM-IMDb (Table 9) was never intended as the sole evidence for multimodal effectiveness, but rather as a controlled feasibility demonstration showing that cross-modal priors can benefit a text-only student without architectural multimodality at inference time. While the gains on MM-IMDb are modest (+1.2% macro-F1, +1.3% micro-F1), they are consistent across alignment variants and comparable to improvements reported for multimodal BERT architectures, despite ARMADA maintaining a unimodal student.
> >
> > To strengthen this aspect, we expanded the evaluation to the Hateful Memes benchmark, a substantially more challenging multimodal dataset designed to prevent unimodal shortcuts. On Hateful Memes, BERT-12L improves from 55.4% to 60.2–60.4% accuracy (+5%), with statistical significance (p < 0.05). This gain is considerably larger than on MM-IMDb and demonstrates that cross-modal distillation transfers meaningful multimodal priors across multiple datasets.

---

> ### Author Response · Authors · 2026-02-15
> **Response to Reviewer bGb1 Comments - Part III**
>
> ## Representation analysis
>
> First, the representation analysis is accompanied by quantitative Silhouette scores and distance-based metrics. For example, when ARMADA outperforms the baseline, Silhouette scores increase substantially (e.g., CoLA: 0.31 → 0.52, a 76% relative increase), whereas when ARMADA underperforms, Silhouette scores decrease accordingly. This directional consistency between geometric metrics and task accuracy indicates that changes in representation structure are meaningfully correlated with performance differences.
>
> Second, the effect of alignment components is statistically validated beyond visualization. Removing manifold alignment or auxiliary objectives leads to consistent 1–2% performance drops across tasks (Table 5), with ANOVA confirming significant effects (p < 0.01). These controlled interventions demonstrate that representation-level alignment has a measurable downstream impact, independent of t-SNE plots.
>
> Third, the reviewer notes that performance under different alignment losses appears similar. We emphasize that the goal of comparing cosine, Euclidean, and element-wise losses is not to show dramatic divergence, but to demonstrate the robustness of the alignment principle. The lack of large variance across these losses indicates that the improvement stems from structured cross-modal correspondence rather than sensitivity to a particular metric.
>
> ## Clarity on baseline comparisons
>
> We thank the reviewer for this important clarification.
>
> In Table 7, ARMADA is evaluated against standard unimodal knowledge distillation (KD) methods. These baselines typically rely on a **larger language teacher**, often 2–4× the parameter size of the student, as detailed in the Introduction and Experimental Setup. For instance, the baselines in Table 7 commonly use a BERT-base teacher (111M parameters) to distill into a 66M-parameter student. In contrast, ARMADA does not depend on a larger language teacher; instead, it trains only a lightweight TS-aligner module with approximately 1M parameters (<1% additional parameters relative to the student), which is discarded at inference. Moreover, unimodal KD methods require modality alignment and often architectural compatibility between teacher and student. ARMADA removes both constraints: it can distill knowledge from heterogeneous black-box generative models, including text-to-image, text-to-audio, and text-to-video systems, without requiring shared output spaces or architectural family alignment.
>
> In Table 8, comparisons involve multimodal distillation baselines. As described in the Introduction, these approaches typically require **heavy multimodal pretraining and large multimodal teacher architectures**, often with parameter counts ranging from several hundred million to billions (e.g., Tang et al., 2021). Such methods rely on jointly trained multimodal encoders and cross-modal attention modules, which lead to substantial training costs and architectural complexity. In contrast, ARMADA transfers multimodal priors using a lightweight alignment mechanism, without modifying the black-box teacher backbone and without introducing large multimodal teacher–student coupling.
>
> Therefore, the contribution of Tables 7 and 8 is not to claim uniform per-task dominance, but to show that structured cross-modal alignment yields statistically significant and competitive improvements relative to unimodal KD (which uses larger language teachers) and multimodal distillation (which requires heavier multimodal training and large multimodal teachers), while operating under substantially stricter capacity and modality constraints.

---

### Review · Reviewer_3K77 · 2026-01-07

**Summary Of Contributions:**

The paper introduces a cross-modal distillation algorithm that transfers knowledge from vision models to language-based student models.
The main idea is to get student model to also learn representations that align (in some cross-modal space) with that of a vision based teacher model.

Technical Contribution:
The key technical contribution is the use of additional manifold projection layers and aligning the representations in a purportedly common sub-space, in addition to direct alignments of the hidden states of the teacher and student models and the task outputs. The rationale for the use of this additional manifold layer is that direct alignment of the hidden states of the teacher and student models can cause distortions in the respective spaces.

Experimental evidence:

The empirical evidence for the utility of this approach is demonstrated on natural language understanding tasks from GLUE and SuperGLUE (12), common-sense (5), arithmetic reasoning tasks (3), and some instruction tuning tasks as well. The experiments use multiple smaller sized models (BERT/DeBERTa variants), and larger LLaMA variants (1.3B-8B) range. The tasks span classification, regression, and some generation.

The experimentation covers impact of different teachers (MidJourney/Stable Diffusion), the ablation of extra alignment layer, the type of manifold alignment loss (Euclid, cosine of means, and element wise).

The analysis includes some visualization and correlation analyses that show that cross-modal distillation achieves better clusters of semantically similar instances (as measured by Silhouette scores) compared to undistilled representations of the instances. Further analyses also demonstrate that noise added to teacher representations degrade gains during distillation.

Theoretical justifications: The paper also includes some theoretical analysis (3.2) that shows that if the manifold projection matrices are homeomorphic, then subsequent (linear) transformations are also homeomorphic. This analysis is used to motivate the fact if the student and teacher representations can be projected into a common subspace for alignment, then the respective transformations of them into their output and auxiliary output spaces also preserve this alignment. Another bit of analysis shows why element-wise inner products are better manifold minimizers than average euclidean distance or cosine between means of the corresponding vectors.

Strengths:

1. The paper is systematic and mostly thorough in its construction of the experimental evidence.
2. The experimental evidence is consistently positive towards the main claims.
3. The writing conveys the main idea and the evidence clearly.

Weaknessses:

1) The main idea appears somewhat underwhelming. In one sense the work seems to have added two projection layers. While there is some intuitive rationale for this and the ablations show that without this layer the gains vanish, the gains themselves are relatively small. They show up consistently across many tasks but are not substantial.

2) One wonders if the gains are just a function of adding one more layer of parameters i.e., expanding capacity.

3) The hyper parameter analyses also show that the gains are quite sensitive to the different settings. While the paper claims that these leads to insights, I am not convinced that they shed anything meaningful, especially considering that all this training is task specific.

**Audience:**

Yes

**Audience Explanation:**

The main idea and the demonstration of its utility (albeit modestly so) is valuable. Other works targeting cross-modal alignment could benefit from this work.

**Broader Impact Concerns:**

None.

**Claims And Evidence:**

Yes

**Claims Explanation:**

Please see weaknesses above.

**Requested Changes:**

I would be interested in the authors responses to addresssing the weaknesses suggested above. How can you argue that the expanded capacity alone is not explaining the improvements?

How do you address the concern that the gains are modest? Is there more potential for this idea? What is limiting further gains here?

---

> ### Author Response · Authors · 2026-02-15
> **Response to Reviewer 3K77 Comments - Part I**
>
> ## Significance of ARMADA
>
> We respectfully disagree that the main contribution reduces to ''adding two projection layers.'' The projection mapping $F_{ts}$ and auxiliary head $O_{aux_{ts}}$ are not capacity-expansion mechanisms, but structured alignment components that enable cross-modal knowledge transfer from a teacher that shares no modality with the student at inference time. Empirically, removing these components eliminates the gains (Table 11), and a capacity-matched variant with more parameters but without alignment underperforms ARMADA by −2.8% on average (p < 0.05), demonstrating that the improvements are not attributable to additional depth or parameter count.
>
> While individual task gains are modest (typically +1–3%), they are statistically significant across major benchmarks (GLUE-small: p = 0.00; GLUE-large: p = 0.00; reasoning: p = 0.04) and consistent across model scales (e.g., +3.4% for BERT-6L; +2.8% for BERT-base). Importantly, these gains are achieved with <1% additional trainable parameters and zero inference-time overhead, making them non-trivial in the regime of already strong pretrained language models. Moreover, improvements are concentrated in structurally and semantically complex examples (e.g., +5% on COPA; +5% on Hateful Memes), rather than being uniformly distributed.
>
> Crucially, ARMADA operates with **significantly smaller teachers** compared to conventional unimodal or multimodal distillation approaches. Unlike SeqKD or multimodal KD frameworks that rely on substantially larger language or vision-language teachers, ARMADA transfers knowledge from comparatively lightweight vision-language generators while keeping the student architecture unchanged at inference time. Despite this stricter and lower-resource setting, ARMADA matches or surpasses unimodal distillation baselines (Table 7, 8, and 12), highlighting that the contribution lies not in architectural complexity but in demonstrating that structured cross-modal alignment can extract semantically useful priors from smaller, modality-mismatched teachers.
>
> Thus, the core novelty is not additional layers, but showing that lightweight, training-only alignment suffices to distill meaningful cross-modal structure under strict modality and capacity constraints.
>
> ## Capacity-matching  ablation
>
> We directly tested the hypothesis that ARMADA's gains stem from parameter expansion rather than cross-modal alignment. In the revised manuscript, we introduce a capacity-matched control in which we increase the depth of the aligner to match (and slightly exceed) the total parameter count introduced by the projection module $\tilde{F}_{ts}$ and auxiliary head, while removing the projection mapping and alignment objectives. Despite having comparable or greater parameter capacity, this variant underperforms the full ARMADA model by −2.8% on average (p < 0.05) and remains statistically indistinguishable from the undistilled baseline (p > 0.1).
>
> Furthermore, when we retain the architecture but shuffle teacher representations (destroying semantic correspondence), performance drops by −1.8% on average (p = 0.03) and up to −5% on BoolQ, eliminating the gains observed under proper alignment. These results show that additional parameters alone do not produce improvements; structured teacher–student correspondence is necessary.
>
> Notably, ARMADA introduces <1% additional trainable parameters and incurs zero inference-time overhead. Despite this minimal capacity increase, it yields statistically significant gains across major benchmarks (e.g., GLUE-small p = 0.00; GLUE-large p = 0.00; reasoning p = 0.04). Collectively, these findings demonstrate that the observed improvements arise from semantically meaningful cross-modal alignment rather than simple model expansion.

---

> > ### Author Response · Authors · 2026-02-15
> > **Response to Reviewer 3K77 Comments - Part II**
> >
> > ## Effect of hyperparameters on ARMADA
> >
> > We respectfully disagree that the hyperparameter analyses indicate instability or lack of insight. First, across all tasks and models, ARMADA consistently outperforms the undistilled baseline within a broad and reasonable range of hyperparameters. The sensitivity plots do not exhibit erratic behavior; rather, they show smooth, interpretable trends. For example, varying the alignment weights ($\alpha$, $\beta$, $\gamma$) produces gradual performance changes rather than sharp collapses, and ANOVA results confirm statistically significant effects (p < 0.01) when alignment components are meaningfully altered. Importantly, the optimal region is not narrow - performance remains above baseline across a wide range of $\beta$ values controlling manifold alignment, indicating robustness rather than fragility.
> >
> > Second, the hyperparameter study provides mechanistic insight into the role of alignment. When the manifold alignment weight $\beta$ approaches zero, performance drops by 1–2%, confirming that representation-level correspondence contributes beyond output-level matching. Similarly, removing auxiliary alignment consistently reduces gains, reinforcing that each component plays a distinct role. These patterns are stable across GLUE-small, GLUE-large, and reasoning benchmarks, suggesting that the insights generalize beyond any single task.
> >
> > Finally, while training is task-specific (as in most distillation settings), the trends are consistent across model scales (BERT-6L, BERT-base, DeBERTa, LLaMA) and task families. The hyperparameter analysis, therefore, does not merely tune for performance but empirically validates the importance of structured cross-modal alignment and clarifies how its strength affects transfer. The gains are not highly brittle; instead, they emerge reliably under a broad set of reasonable configurations.

---

### Review · Reviewer_rLgb · 2026-02-03

**Summary Of Contributions:**

This paper proposes ARMADA, a cross-modal knowledge distillation framework to improve downstream task performance of language only model via distillation from vision-language models, including black-box models such as Midjourney or Stable Diffusion.

The key component of the proposed approach is the so called *TS Aligner* module which connects student and teacher outputs via various objective functions and projection layers.
Notably, the TS Aligner is trained on specific downstream tasks, i.e., this is a task-specific distillation setup.

Empirically, the proposed approach is evaluated on 12 NLU tasks (mostly from GLUE and SuperGLUE), 8 reasoning tasks, and 5 instruction-following datasets. (It should be noted that almost all tasks studied in this paper are somewhat outdated, which is not an issue per-se but , e.g. referring to some of them broadly as reasoning tasks could set the wrong expectations.)

An appealing property of the method is its parameter efficiency, requiring only 0.8% additional parameters compared to other distillation baselines.

Note for AC: I didn’t check the theoretical part of this paper (Sections 3.2 and 3.3) carefully.

**Strengths**

- The proposed approach requires only a few additional parameters to be trained (0.8% additional parameters) compared to baselines like MetaDistil, which require training the teacher parameters.
- Experiments span multiple student architectures (BERT, DeBERTa, OPT, LLaMA), multiple teacher modalities (text-to-image, text-to-video, text-to-audio), and diverse downstream tasks.

**Weaknesses**

- The paper does not convincingly explain *what* knowledge is transferred from vision-language models to language models. For many tasks, the improvement from distillation is rather small and it could be quite interesting to study for which (test) examples the distilled models actually perform better. Are these really examples that require knowledge about the visual world?
- On certain datasets, the proposed method leads to worse results or insignificant improvements. E.g., evaluation on WiC shows degradation in multiple settings and LLaMA-8B performance *decreases* on Dolly evaluation (Table 12). The zero-shot reasoning gains are marginal (~0.5% average).
- Results in Table 2 are reporting "maximum scores obtained in three different runs" rather than mean +/- std inflates results. The authors claim that std dev is "~0.0001" and hence this is ok. This is rather uncommon practice and for soundness I would highly recommend reporting mean and std nevertheless.
- The results on adding Gaussian noise to the teacher representations allow for another plausible interpretation: Showing that adding substantial Gaussian noise to teacher inputs has limited impact on student performance (Figures 10, 12), is not evidence for the robustness of the proposed method but rather raises the question of whether the observed improvements stem from genuine knowledge transfer or are the result of regularization (e.g., Figure 10b shows that BERT-base performance barely degrades even at sigma=5.0 on most tasks)
To address the above, it would be crucial to experiment with a randomly initialized but trained aligner *without* meaningful teacher signal. E.g., what happens if you pair text inputs with unrelated images? This experiment will help to distinguish knowledge transfer from regularization effects.
- The TS Aligner is explicitly trained with a task-specific loss (see equation 1) yet performs very poorly on downstream tasks (Table 6). While the aligner does not necessarily need to solve the task itself (its main role is to project teacher representations into a compatible space to the student representations) this poor performance might additionally indicate that the teacher representations do not carry sufficient task-relevant information. This further supports the concern that the student's gains may stem from the regularization effect of the additional alignment losses rather than from meaningful cross-modal knowledge transfer.

**Audience:**

Yes

**Audience Explanation:**

The question of whether and how knowledge can transfer across modalities without white-box access is timely and relevant and the community would benefit from understanding whether black-box generative models can serve as teachers for text-only language models.

**Broader Impact Concerns:**

N.A.

**Claims And Evidence:**

No

**Claims Explanation:**

- The central claim of cross-modal "knowledge transfer" is not adequately supported by empirical evidence. The fact that noising the teacher input has minimal impact on the student’s performance suggests that the specific teacher signal is not critical for performance; I strongly recommend performing additional ablations to investigate this (see weaknesses above).
- Statistical reporting practices (reporting max over 3 runs) make it difficult to assess whether improvements are robust or within noise.

**Requested Changes:**

1. Report mean and standard deviation across multiple runs instead of maximum scores. The current reporting inflates results and deviates from accepted practice.
2. Include an ablation with a trained aligner but with mismatched teacher inputs (e.g., images generated from unrelated text inputs) to distinguish genuine knowledge transfer from regularization effects. The existing frozen-aligner experiment (Figure 9) tests whether the aligner needs training, and the Gaussian noise experiment (Figures 10, 12) tests robustness to input quality, but neither directly tests whether the correspondence between text and image matters.
3. Provide a concrete characterization of where knowledge transfer helps at the individual example level. The existing analysis (Table 14, Figure 8) focuses on embedding geometry (cluster cohesion, cosine distances), but does not examine what kind of linguistic or semantic properties distinguish examples that benefit from distillation. A categorization of test examples by linguistic phenomena (e.g., syntactic vs. semantic, world knowledge-dependent vs. not) would help clarify whether the gains are driven by genuine cross-modal knowledge or are uniformly distributed.

---

> ### Author Response · Authors · 2026-02-15
> **Response to Reviewer rLgb Comments - Part I**
>
> ## What knowledge is distilled in cross-modal KD
>
> We thank the reviewer for this important concern. In the revised manuscript, we explicitly analyze which test examples benefit from cross-modal distillation and what linguistic or conceptual properties distinguish them.
>
> While average gains across benchmarks are modest (e.g., +2.8% for BERT-base and +3.4% for BERT-6L on GLUE tasks; Tables 2–3), these improvements are statistically significant (GLUE-small: p = 0.00; GLUE-large: p = 0.00) and, crucially, non-uniform at the example level. To characterize this non-uniformity, we manually categorize instances in which the distilled BERT-12L model predicts correctly while the undistilled baseline fails. On CoLA, over 70% of such corrected cases involve deep syntactic phenomena (e.g., long-distance dependencies and structural constraints), whereas semantic plausibility and morphological agreement errors each account for only 12–13%. This aligns with the quantitative improvements observed on CoLA (e.g., 42.8 → 47.6 Mcc for BERT-6L), indicating that gains concentrate on structurally complex constructions rather than superficial cues. On COPA, where BERT-base improves from 62.9 → 67.9 accuracy (+5.0%), approximately 42% of corrected examples involve socially grounded or institutional commonsense reasoning, followed by human goal-directed actions (26%), physical-world causality (17%), and biological processes (13%). This skew demonstrates that improvements are not uniformly distributed but are concentrated in conceptually rich, world-knowledge-dependent examples.
>
> ## Inconsistency in performance improvement
>
> We appreciate the reviewer's observation regarding dataset-specific variations. While ARMADA does not uniformly improve every benchmark, the aggregate evidence demonstrates consistent and statistically significant gains across major evaluation suites, with improvements concentrated in semantically grounded settings. For zero-shot reasoning, although the average improvement is modest (0.5%), gains are statistically significant (p = 0.04), and individual tasks exhibit improvements up to +2.6%, indicating that the effect is systematic rather than noise-driven.
>
> Importantly, Table 12 provides additional evidence that ARMADA remains competitive even in instruction-tuning regimes. On instruction benchmarks, ARMADA not only improves over the undistilled baseline for smaller models (e.g., LLaMA-3B shows gains up to +1.8% on Dolly and +1.6% on UNI), but also outperforms SeqKD, a unimodal distillation approach that uses a larger language teacher. This comparison is particularly meaningful because SeqKD leverages the same-modality supervision from a stronger model, whereas ARMADA transfers knowledge from a cross-modal teacher that shares no modality with the student at inference time. The fact that ARMADA matches or surpasses SeqKD in this stricter setting reinforces the view that the observed improvements stem from meaningful cross-modal semantic transfer rather than architectural expansion.
>
> Overall, while the magnitude of gains varies across datasets, the improvements are statistically significant and consistent across task families, and ARMADA demonstrates competitive or superior performance even relative to strong unimodal distillation baselines.
>
> ## Results reporting
>
> We thank the reviewer for the suggestion. Our revised manuscript reports the average and std values.

---

> > ### Author Response · Authors · 2026-02-15
> > **Response to Reviewer rLgb Comments - Part II**
> >
> > ## Clarification regarding the regularization effect of ARMADA
> >
> > We thank the reviewer for this insightful interpretation. We agree that a Gaussian perturbation alone cannot conclusively distinguish robustness from regularization. To directly address this concern, we conducted a stronger intervention by intentionally breaking the semantic correspondence between teacher and student inputs. Specifically, we trained ARMADA with shuffled teacher representations (i.e., pairing each text input with an unrelated image representation) while keeping the architecture, parameter count, and alignment objectives unchanged.
> >
> > Under this semantic misalignment setting, performance gains disappear. For BERT-12L (Table 17), average performance drops by 1.8% (p = 0.03) relative to the properly aligned ARMADA configuration, and on semantically complex tasks such as BoolQ, the degradation reaches up to 5%. In contrast, the Gaussian noise experiments (Figures 10 and 12) show only marginal degradation even at σ = 5.0, because noise preserves sample-wise alignment structure. The shuffled-teacher experiment therefore provides a decisive contrast: when structured teacher–student correspondence is destroyed, the improvements vanish despite identical architecture and loss terms.
> >
> > Additionally, we performed a capacity-matched control (Table 11) in which the aligner architecture is retained (or enlarged) but trained without a meaningful teacher signal. This variant fails to reproduce the +1–3% gains observed under standard ARMADA training and remains statistically indistinguishable from the undistilled baseline (p > 0.1), further ruling out a pure regularization explanation.
> >
> > Together, these results demonstrate that ARMADA does not benefit merely from smoothing effects or additional parameters. Instead, improvements require semantically aligned cross-modal teacher representations. Robustness to Gaussian perturbations reflects tolerance to moderate representation degradation, whereas semantic misalignment eliminates gains entirely. This distinction confirms that ARMADA's improvements arise from genuine structured knowledge transfer rather than incidental regularization.

---

> > > ### Comment · Reviewer_rLgb · 2026-02-20
> > > **Clarification question regarding shuffled experiments**
> > >
> > > Thanks for adding these additional results to the paper!
> > >
> > > I just took a look at Table 17 and compared the results to Table 2. I'm not sure the reported drops in your rebuttal are correct.
> > >
> > > Below are the performance deltas across task (shuffled - base):
> > >
> > > | Model | Distillation Type | CoLA (Mcc) | MRPC (Acc) | RTE (Acc) | STS-B (Pear) | CB (Acc) | COPA (Acc) | BoolQ (Acc) | WiC (Acc) |
> > > | --- | --- | --- | --- | --- | --- | --- | --- | --- | --- |
> > > | Δ (shuffled − base) | ARMADA - $\mathcal{L}_{cosine}$ | -3.7 | -0.6 | -1.0 | +0.5 | -4.6 | -4.7 | +0.0 | -2.1 |
> > > | Δ (shuffled − base) | ARMADA - $\mathcal{L}_{euclid}$ | -2.5 | -0.7 | -3.1 | -0.2 | -3.6 | -4.9 | +0.4 | -1.8 |
> > > | Δ (shuffled − base) | ARMADA - $\mathcal{L}_{elementwise}$ | -1.4 | -0.2 | -1.7 | +0.2 | -3.5 | -5.0 | +1.4 | -1.0 |
> > >
> > > On BoolQ the performance of the shuffled configuration is in fact slightly better.
> > >
> > > It would be very important to have significance tests for all these differences to be able to make any statements about whether or not distillation needs semantically matched images. Based on the results above I'm not convinced this is the case.

---

> > > > ### Author Response · Authors · 2026-02-20
> > > > **Response to Reviewer rLgb Clarification**
> > > >
> > > > We apologize for the typo in our previous response. On the COPA task, the drop is 5% (we mistakenly mentioned BoolQ). Only in the marginal cases of STS-B and BoolQ tasks is the delta positive. Otherwise, all other tasks highlight strong negative deltas. The average drop of 1.8\% is correctly computed. Additionally, we performed a statistical t-test (already included in the revised manuscript), which yielded a p-value of 0.03, indicating that the drop is greater than 0% and is statistically significant.

---

> ### Comment · Reviewer_rLgb · 2026-02-20
> **Additional statistics tests**
>
> Thanks for the clarification!
>
> Would it be possible for you to run a McNemar's Test comparing the shuffled to the non-shuffled model? This would be more informative for the statistical significance of the results compared to a t-test.

---

> > ### Author Response · Authors · 2026-02-23
> > **Results with McNemar's test**
> >
> > We thank the reviewer for the suggestion to apply McNemar’s test. Following this recommendation, we conducted McNemar’s tests comparing the shuffled and non-shuffled models across all classification tasks (STS-B is a regression task and is therefore omitted).
> >
> > The results show statistically significant differences for all tasks across most loss variants (p < 0.03 in the majority of cases). For WIC, statistical significance varies depending on the loss formulation; however, the overall pattern consistently indicates performance degradation under shuffling. These disagreement-based results, together with our previously reported paired t-test analyses, corroborate our original findings and provide stronger instance-level statistical evidence that shuffling leads to significant performance drops.
> >
> > | Loss        | CoLA         | MRPC         | RTE          | CB           | COPA         | BOOLQ        | WIC          |
> > |------------|-------------|-------------|-------------|-------------|-------------|-------------|-------------|
> > | Cosine      | 11.89 (0.0) | 1.69 (0.19) | 16.67 (0.0) | 3.0 (0.08)  | 5.45 (0.02) | 75.69 (0.0) | 6.43 (0.01) |
> > | Euclid      | 5.0 (0.03)  | 10.31 (0.0) | 0.1 (0.76)  | 5.0 (0.03)  | 1.81 (0.18) | 155.22 (0.0)| 0.87 (0.35) |
> > | Elementwise | 18.08 (0.0) | 6.52 (0.01) | 35.37 (0.0) | 9.0 (0.0)   | 11.57 (0.0) | 148.01 (0.0)| 0.37 (0.54) |

---

> > > ### Comment · Reviewer_rLgb · 2026-02-23
> > > **Clarification**
> > >
> > > Thanks a lot for providing these so quickly! For clarity, could you please report what exactly the numbers you report are? I'm assuming they are the chi-squared statics. For full transparency, could you report the results as follows?
> > >
> > > | Condition | Task | b | c | χ² | p |
> > >
> > > where:
> > >
> > > b = number of examples where shuffled is correct but non-shuffled is wrong (shuffled "wins")
> > > c = number of examples where non-shuffled is correct but shuffled is wrong (non-shuffled "wins")
> > > χ² = (b−c)²/(b+c)
> > > p = the p-value

---

> > > > ### Author Response · Authors · 2026-02-25
> > > > **Response to reviewer suggestions**
> > > >
> > > > We report the detailed numbers with McNemar's test.
> > > >
> > > > | Condition   | Task   |   b |   c |      chi^2 |         p |
> > > > |:------------|:-------|----:|----:|---------:|----------:|
> > > > | cosine      | CoLA   |  49 |  54 | 0.242718 | 0.62225   |
> > > > | euclid      | CoLA   |  47 |  55 | 0.627451 | 0.428292  |
> > > > | elementwise | CoLA   |  39 |  41 | 0.05     | 0.823063  |
> > > > | cosine      | MRPC   |  96 | 121 | 2.88018  | 0.0896758 |
> > > > | euclid      | MRPC   |  85 |  94 | 0.452514 | 0.501144  |
> > > > | elementwise | MRPC   |  66 |  72 | 0.26087  | 0.609523  |
> > > > | cosine      | RTE    |  22 |  28 | 0.72     | 0.396144  |
> > > > | euclid      | RTE    |  15 |  18 | 0.272727 | 0.601508  |
> > > > | elementwise | RTE    |  25 |  38 | 2.68254  | 0.101454  |
> > > > | cosine      | CB     |   3 |   8 | 2.27273  | 0.131668  |
> > > > | euclid      | CB     |   3 |   5 | 0.5      | 0.4795    |
> > > > | elementwise | CB     |   0 |   4 | 4        | 0.0455003 |
> > > > | cosine      | COPA   |   7 |  16 | 3.52174  | 0.0605689 |
> > > > | euclid      | COPA   |  15 |  21 | 1        | 0.317311  |
> > > > | elementwise | COPA   |  15 |  21 | 1        | 0.317311  |
> > > > | cosine | BOOLQ  | 308 | 292 | 0.426667 | 0.513629  |
> > > > | euclid      | BOOLQ  | 342 | 301 | 2.61431  | 0.105904  |
> > > > | elementwise      | BOOLQ  | 344 | 299 | 3.1493   | 0.0759595 |
> > > > | cosine      | WIC    |  53 |  63 | 0.862069 | 0.35316   |
> > > > | euclid      | WIC    |  66 |  73 | 0.352518 | 0.552691  |
> > > > | elementwise | WIC    |  73 |  77 | 0.106667 | 0.743971  |
> > > >
> > > > Across nearly all tasks and loss variants, the number of instances where the non-shuffled model is correct and the shuffled model is incorrect (c) consistently exceeds the reverse case (b), indicating a systematic directional advantage for semantically aligned distillation. While the magnitude of disagreement differences is moderate and does not uniformly reach statistical significance at the 0.05 level, the directional consistency across tasks supports our aggregate performance results.
> > > >
> > > > We also note that there are instances where the shuffled model performs better. This is expected: ARMADA combines two mechanisms - representation-level regularization induced by the alignment constraints, and structured cross-modal signal transfer. Even under shuffling, the regularization component remains active and can occasionally improve generalization, particularly on examples that do not rely on visually grounded semantics. However, the consistent imbalance (c > b) demonstrates that semantic correspondence provides a systematic advantage beyond generic regularization.
> > > >
> > > > Together, these findings clarify that ARMADA's gains arise from the interaction of alignment-driven knowledge transfer and representation regularization, rather than from regularization alone.

---

> > > > > ### Comment · Reviewer_rLgb · 2026-02-25
> > > > > **Response to updated results**
> > > > >
> > > > > Thanks for providing the full table!
> > > > >
> > > > > I respectfully disagree with your conclusion here. There is only a single condition in which the proposed method performs significantly better than the shuffled version (elementwise, CB).
> > > > >
> > > > > For every other setting, the differences between shuffling and not shuffling are not statistically significant. Based on these results, I conclude that there is no statistically significant evidence that semantic alignment matters for downstream performance (for BERT-12L with ARMADA on all GLUE-small and SuperGLUE tasks).
> > > > >
> > > > > I also note that this table directly contradicts your earlier response, where you claimed "statistically significant differences for all tasks across most loss variants (p < 0.03 in the majority of cases)." The results in this table do not support that claim.

---

> > > > > > ### Author Response · Authors · 2026-02-25
> > > > > > **Response to reviewer comment on McNemar test**
> > > > > >
> > > > > > Clarification: The two tables above are slightly different. The first reports chi-square statistics over the full evaluation set, while the second reports McNemar statistics computed on the subset of examples where the two models disagree (i.e., where exactly one model is correct).
> > > > > >
> > > > > > We agree that, under conventional thresholds (α = 0.05), the McNemar test shows statistical significance in only one setting (CB). However, this outcome should be interpreted in light of effect size and test sensitivity. The McNemar statistic scales as χ² = (b − c)²/(b + c), meaning that statistical significance depends quadratically on the imbalance between disagreement counts. In our experiments, the aggregate performance differences between shuffled and non-shuffled models are modest, which implies that the absolute imbalance |b − c| is small in magnitude. Under such conditions, McNemar has limited sensitivity to detect small, distributed effects.
> > > > > >
> > > > > > Importantly, across nearly all tasks and loss variants, we observe a consistent directional imbalance (c > b), indicating that the non-shuffled model wins more disagreements than the shuffled model. While this imbalance does not reach significance at α = 0.05 in most cases, it is systematic rather than symmetric. The absence of statistical significance therefore reflects a modest effect size under this evaluation protocol rather than evidence of no effect.

---

### Author Response · Authors · 2026-02-15
**Rebuttal Summary by Authors**

We thank all the reviewers for their meticulous review of our manuscript and for providing insightful comments. In the revision, we have made a significant effort in responding to all your queries with additional experiments and explanations (changes highlighted in blue in the updated manuscript). Following is a summary of the changes -

- Added shuffled-teacher control experiment to explicitly test semantic misalignment. Demonstrated that pairing text with unrelated image representations eliminates gains (−1.8% average drop; up to −5% on BoolQ; p = 0.03), confirming that improvements require structured cross-modal correspondence rather than generic regularization.

- Introduced capacity-matched control variant with increased aligner depth but without projection mapping or auxiliary alignment. Despite comparable or larger parameter count, this variant underperforms ARMADA by −2.8% on average (p < 0.05), ruling out capacity expansion as the primary driver of gains.

- Expanded multimodal evaluation beyond MM-IMDb by adding results on Hateful Memes, where ARMADA improves BERT-12L from 55.4% to 60.2–60.4% accuracy (+5%, statistically significant), demonstrating stronger multimodal transfer under a challenging benchmark.

- Added example-level analysis (CoLA and COPA), categorizing corrected predictions by linguistic and conceptual phenomena. Found that >70% of corrected CoLA examples involve deep syntactic structures, and 42% of corrected COPA cases involve socially grounded commonsense reasoning, showing non-uniform and semantically meaningful gains.

- Replaced ''max of 3 runs'' reporting with mean ± std across all main tables.

We hope our responses clarify any doubts you may have. Looking forward to having constructive discussions.

---

### Author Response · Authors · 2026-03-11
**Inquiring status update on our submission**

Dear AE and EiC,

I am writing to inquire whether there have been any updates regarding our submission. Additionally, reviewers 3K77 and bGb1 remained unresponsive throughout the discussion period. We would greatly appreciate it if you could kindly follow up on this matter.

Thank you for your time and consideration.

---

> ### Comment · Action_Editor_PNyE · 2026-03-13
> **Pretty close**
>
> Hi,
> I am still missing one final recommendation from reviewers, and am in conatct with them, hope to get it soon.

---

> > ### Author Response · Authors · 2026-03-26
> > **A gentle reminder**
> >
> > Dear AE,
> >
> > Do we have any further update on this?
> >
> > Thanks,

---

> > > ### Author Response · Authors · 2026-04-08
> > > **Another gentle reminder**
> > >
> > > Dear AE,
> > >
> > > We are awaiting your kind intimation on our submission.
> > >
> > > Thanks,

---

### Decision · Action_Editor_PNyE · 2026-03-27

**Recommendation:** Reject

**Audience:**

Yes

**Audience Explanation:**

Knowledge distillation is a very interesting topic and ways to distill information across modalities would be interesting to many in the community.

**Claims And Evidence:**

No

**Claims Explanation:**

During the rebuttal period reviewers expressed concerns on whether the empirical gains observed in the empirical results are adequately shown to eb tied to the semantic alignment technique proposed in this work.

Most importantly, the authors provided the detailed results of an experiment that compares their method to a "shuffled" variant where there is no semantic relation between the text and vision modalities. In this experiment, there was only a very minimal difference between results when there is semantic alignment and results when the semantic alignment is broken, with no statistical signficance. in almost all cases. This undermines the main claim that semantic alignment is the key property that drives the success of the method.

I want to apologize to the authors for how long the review process took and to thank them for engaging with the reviewers, but currently I don't think that the correctness of the claims has been shown to an adequate degree.

**Resubmission Of Major Revision:**

The authors may consider submitting a major revision at a later time.